# A Narrative Review on the Neurocognitive Profiles in Eating Disorders and Higher Weight Individuals: Insights for Targeted Interventions

**DOI:** 10.3390/nu16244418

**Published:** 2024-12-23

**Authors:** Isabel Krug, An Binh Dang, Evonne Lu, Wenn Lynn Ooi, Jade Portingale, Stephanie Miles

**Affiliations:** 1Melbourne School of Psychological Sciences, The University of Melbourne, Melbourne, VIC 3010, Australia; dangb1@student.unimelb.edu.au (A.B.D.); wennlynn.ooi@gmail.com (W.L.O.); jportingale@student.unimelb.edu.au (J.P.); 2Monash Faculty of Medicine, Nursing and Health Sciences, Monash University, Melbourne, VIC 3800, Australia; eluu0007@student.monash.edu; 3Orygen, Parkville, VIC 3052, Australia; stephanie.miles@orygen.org.au; 4Centre for Youth Mental Health, The University of Melbourne, Melbourne, VIC 3010, Australia; 5Department of Psychological Sciences, Swinburne University of Technology, Hawthorn, VIC 3122, Australia

**Keywords:** anorexia nervosa, bulimia nervosa, binge eating disorder, otherwise specified feeding or eating disorders, higher weight, neurocognitive profile, executive functioning

## Abstract

**Background/Objectives**: Recent research has increasingly explored the cognitive processes underlying eating disorders (EDs), including anorexia nervosa (AN), bulimia nervosa (BN), binge eating disorder (BED), other specified feeding or eating disorders (OSFEDs), and individuals with higher weight (HW). This critical narrative review focuses on neurocognitive findings derived from mainly experimental tasks to provide a detailed understanding of cognitive functioning across these groups. Where experimental data are lacking, we draw on self-report measures and neuroimaging findings to offer supplementary insights. **Method:** A search of major databases that prioritized meta-analyses and recent publications (last 10 years) was conducted. Using comprehensive search terms related to EDs, HW, and neurocognition, eligible studies focused on human neurocognitive outcomes (e.g., cognitive flexibility, attentional bias, etc.) published in English were selected. **Results:** We found that some neurocognitive characteristics, such as cognitive rigidity, impulsivity, emotion processing difficulties, and dysregulated reward processing, appear transdiagnostic, spanning multiple ED subtypes and HW populations. We also revealed neurocognitive features specific to ED subtypes and HW. For instance, individuals with AN demonstrate an enhanced focus on detail, and BN and BED are characterized by a pronounced attentional bias toward food-related stimuli. In individuals with HW, cognitive processes underpin behaviours associated with overeating and weight gain. **Conclusions:** These findings highlight the critical importance of understanding both the unique and shared neurocognitive patterns across ED subtypes and HW populations. By identifying transdiagnostic factors, such as cognitive rigidity and reward processing, alongside ED subtype/HW-specific vulnerabilities, researchers and clinicians can develop more nuanced, evidence-based interventions that address the core mechanisms driving disordered eating behaviours.

## 1. Introduction

Neurocognitive characteristics have been increasingly recognized as significant factors in the development and maintenance of eating disorders (EDs). Deficits in executive functioning and neurocognitive symptoms such as impulsivity [1], cognitive inflexibility [2], and poor emotion regulation [3] have been associated with several EDs. However, it remains unknown if these neurocognitive difficulties are intrinsic to EDs and part of the illness or are secondary consequences of malnutrition and changes in weight.

While EDs, including anorexia nervosa (AN), bulimia nervosa (BN), binge eating disorder (BED), and other specified feeding or eating disorders (OSFEDs), involve distinct behavioural patterns, they all share common underlying disturbances related to eating, body image, and weight [4,5]. Further, when examining EDs, considering people with a higher weight (HW) can provide additional insights—not as an ED, but as a condition situated at one end of the weight spectrum.

In this review, the term HW will refer to individuals whose body weight or body mass index (BMI) exceeds what health guidelines consider the “normal” range. Specifically, the term HW includes individuals with BMIs classified as overweight (BMI 25–29.9) or obese (BMI ≥ 30) according to the criteria set forth by the World Health Organization [6]. While HW is not classified as an ED, it is often associated with maladaptive eating behaviours that overlap with the characteristics of EDs, particularly BED and BN. For instance, one previous study found that lifetime obesity was as high as 87% in a treatment-seeking BED sample [7]. HW individuals also often exhibit neurocognitive and emotional dysfunctions like those seen in ED populations, including impaired cognitive flexibility [8], reward processing [9], heightened impulsivity [10], and emotion processing difficulties [11].

AN and HW represent extremes of weight dysregulation, and integrating HW into a broader examination of neurocognitive functioning is both relevant and necessary for understanding the full scope of weight-related problems. While various studies assessing HW have referred to their sample as “suffering from obesity”, we use the terminology of HW throughout the review. This term is preferred in scientific and clinical contexts to reduce stigma and bias, promoting a more neutral, person-centred approach that respects body diversity and lessens the emphasis on weight as a sole health measure [12].

Furthermore, it is important to acknowledge potential biases in how HW is categorized in research. BMI, while widely used, does not account for body composition, distribution of fat, or metabolic health, potentially oversimplifying a complex construct [12,13]. Cultural and medical biases in the HW terminology, such as framing HW primarily as a pathological state, can also shape the focus of research and influence the interpretation of findings [13,14]. These biases may perpetuate stigmatizing narratives, overlook protective or neutral factors, and obscure the heterogeneity of HW populations. A critical examination of these issues is essential to accurately interpret findings and ensure that the inclusion of HW populations in ED research advances understanding without reinforcing stereotypes or over-simplifications.

This review will examine neurocognitive domains in different weight categories, including underweight (AN), “normal” weight (BN, OSFED), and HW (BED, HW) by primarily focusing on data from experimental tasks. Where experimental data are limited or unavailable, self-report data will also be considered. By examining neurocognitive profiles across EDs and HW, this review seeks to disentangle whether the observed cognitive patterns are a consequence of extreme weight states (underweight in AN or overweight/HW) or reflect core features of the disorders/conditions themselves. Understanding these distinctions is crucial for identifying the underlying mechanisms driving these conditions and for developing interventions that target the specific neurocognitive processes contributing to their onset and maintenance.

Avoidant/restrictive food intake disorder (ARFID) was not included in this review because it differs from traditional EDs like AN, BN, BED, OSFED, and HW in terms of their clinical presentation. ARFID is characterized by food avoidance based on sensory characteristics rather than body image concerns or binge–purge behaviours [4,5]. Therefore, its exclusion allows for a more focused examination of the neurocognitive characteristics of ED subtypes and HW populations who present with a more similar clinical presentation.

## 2. Method

Narrative reviews do not follow the same structured approach as systematic reviews using reporting tools such as the Preferred Reporting Items for Systematic Reviews and Meta-Analyses (PRISMA) guidelines [15]. Instead, they allow for a more flexible and qualitative synthesis of the existing literature without strict inclusion/exclusion criteria, with the aim of focusing on interpreting and synthesizing findings in a more narrative manner.

This narrative review was conducted in three steps: search execution, abstract and full-text review, and write-up and discussion of the main findings. We conducted a comprehensive literature search using major scientific databases, including PubMed, PsycINFO, Web of Science, and Google Scholar. Using these databases, a systematic search strategy was employed to identify relevant studies. The search covered mainly publications from the last 10 years to ensure the inclusion of the most current research, but older articles were also considered if they were key to the literature. The final search, completed in November 2024, included international articles, online reports, and electronic books. Where possible, meta-analyses or other reviews were prioritized to summarize the literature, with specific articles selected to provide illustrative examples or to address gaps where reviews were unavailable. Keywords including “eating disorders”, “higher weight”, “obesity”, “anorexia nervosa”, “bulimia nervosa”, “binge eating disorder”, “OSFED”, “atypical anorexia nervosa”, “purging disorder”, “night eating syndrome”, “subthreshold bulimia/binge eating disorder” to capture EDs/HW were combined with the following neurocognitive search terms: “cognitive flexibility/inflexibility”, “central coherence”, “attentional bias”, “reward processing”, “inhibitory control”, “impulsivity”, “reward processing”, and “emotion regulation/processing”. Studies included in this review met the following criteria:Human studies, with a cross-sectional, longitudinal, or experimental design.Research focused on neurocognitive/imaging outcomes related to EDs or HW.Articles published in English.

Studies were excluded if they did not specify neurocognitive/imaging outcomes, were animal studies, lacked primary data, or were not published in English. The initial selection involved screening titles and abstracts to exclude studies that did not meet the criteria. Full-text articles of relevant studies were reviewed for detailed analysis. Reference lists from selected studies were also reviewed for additional sources.

## 3. Anorexia Nervosa

According to the DSM-5, AN is an ED characterized by a persistent restriction of energy intake, an intense fear of gaining weight or becoming fat, and a distorted body image. AN results in a significantly low body weight for the individual’s age, sex, and developmental trajectory. It has two primary subtypes: the restrictive subtype (AN-R), and the binge–purging subtype (AN-BP), which is characterized by cycles of binge eating followed by purging behaviours to prevent weight gain [4,5]. AN is associated with numerous severe complications, including electrolyte imbalances, reduced bone density, cognitive impairments, and psychiatric comorbidities, notably anxiety, depression, and obsessive–compulsive traits [16,17].

Beyond the disorder’s characteristic cognitive and behavioural distortions concerning food and body image, AN is also linked to neurocognitive deficits in areas such as cognitive flexibility, decision-making, and central coherence [18]. These impairments may be intrinsic to AN or could result from malnutrition and low BMI [19]. While the role of starvation in AN remains unclear, certain neurocognitive functions appear preserved, with general intelligence typically within normal ranges [20]. This preservation supports treatment engagement and recovery, underscoring the importance of addressing specific deficits while leveraging premorbid cognitive strengths.

### 3.1. Cognitive Flexibility

Cognitive flexibility refers to the ability to adapt to changing task demands or environmental shifts by adjusting behaviours and/or thoughts [2]. One prominent construct within cognitive flexibility is set shifting, or the capacity to switch between different mental sets or rules. In AN, cognitive inflexibility has been proposed as a mechanism that may perpetuate symptoms and contribute to treatment resistance [2]. Poor cognitive flexibility is reflected in behaviours such as compulsive weight control, restricted eating, ritualized food consumption, resistance to adopting new habits [1], and difficulties in social interactions [21]. While some studies indicate that individuals with AN exhibit poorer cognitive flexibility compared to healthy controls (HCs) [22,23,24], other research has reported no significant differences between these groups in cognitive flexibility tasks [25,26,27].

A systematic review by Miles et al. (2020) [2] highlights these mixed findings, showing that adolescents with AN perform comparably to HCs on cognitive flexibility tasks, whereas adults with AN often demonstrate poorer performance on perceptual-based cognitive flexibility tasks, yet no significant group difference was supported for verbal cognitive flexibility tasks. Miles et al. (2020) [2] noted that the 69 studies included in their review used over 30 different measures of cognitive flexibility, creating inconsistencies that hinder synthesis and may contribute to conflicting findings. The review further notes variability in findings for weight-restored and fully recovered AN samples, suggesting that while some aspects of cognitive flexibility may improve, certain deficits may persist even after recovery or weight restoration [2].

Other research has explored the relationship between cognitive flexibility and BMI in AN as BMI is suggested as a severity indicator for AN per both the DSM-5 [4,5] and ICD-11 [28]. A recent systematic review found no association between cognitive flexibility and BMI [20], and this finding was supported by subsequent studies [29,30]. These findings suggest that cognitive inflexibility in AN may not be merely a consequence of low weight or starvation but may instead represent a core feature of the disorder.

Understanding cognitive inflexibility as a central aspect of AN points to promising clinical interventions. Targeting this rigidity through cognitive training such as Cognitive Remediation Therapy (CRT) [31], increasing awareness of inflexible thought patterns, and designing behavioural exercises to improve set-shifting abilities may help address key maintenance factors and support long-term recovery.

### 3.2. Central Coherence

Central coherence denotes to the ability to grasp complex stimuli or information, often described as “big picture” thinking, and weak central coherence is typified by an excessive focus on small details. A systematic review and meta-analysis conducted by Keegan et al. (2021) [32] concluded that participants with AN exhibited significantly poorer central coherence compared to HCs. Regarding the relationship between central coherence and BMI, studies have produced varied results [20]. For example, López et al. (2008) [33] found in a sample of women with AN that there was a negative association between BMI and errors on the Homograph Reading Task, a measure of central coherence, yet there were no other significant correlations between BMI and the outcomes of other central coherence tasks included in the study. Conversely, Herbrich et al. (2018) [34] observed a negative correlation between BMI and performance on the Group Embedded Figures Test in participants with AN-BP, but not in the AN-R group.

While most findings suggest deficits in central coherence among those with AN, the findings remain somewhat inconsistent. This variability is partly due to the heterogeneity of studies and methodological differences that need to be considered. Various tests have been used to measure central coherence, raising questions about the comparability of results [2]. Different cognitive tests vary in complexity and often target multiple cognitive functions. For instance, the Homograph Reading Task focuses on linguistic and contextual coherence, whereas the Group Embedded Figures Test assesses perceptual and visual-spatial aspects, each providing distinct insights into different components of central coherence.

In summary, although more research is needed to clarify the relationship between BMI and central coherence in AN, the current evidence points to central coherence deficits among those with AN. A deeper understanding of neuropsychological functioning in AN, particularly its link to neural mechanisms and symptoms, could inform more effective treatments.

### 3.3. Attentional Bias

Attentional bias, the tendency to prioritize processing certain stimuli over others, has been extensively studied in AN. Research investigating attentional processes in AN employs diverse methods, including variations in measurement procedures, types of stimuli (e.g., words or images), threat content, and outcome measures. Despite this methodological diversity, findings consistently show heightened attentional bias toward body shape, weight, and food-related stimuli in individuals with AN compared to HCs [12,35]. Specifically, the Stroop Task and Dot-Probe Task, in combination with eye-tracking paradigms, are commonly used to assess this bias. Eye-tracking research demonstrates that individuals with AN initially display an attentional bias towards food cues that is similar to HCs, but rather than sustaining this attention, individuals with AN quickly switch to attentional avoidance [36]. Werthmann et al. (2019) [36] further found that avoidance was stronger for high-calorie food cues, especially in adults with longer AN illness durations compared to adolescents. This avoidance behaviour tends to become more entrenched as the illness progresses, potentially maintaining the fear of food or restrictive eating behaviours [37]. 

In terms of BMI, while some work has found no association between BMI and attentional bias [38], other research has revealed that those with a low BMI are more likely to pay attention to high-calorie foods [36]. Future studies should further investigate the relationship between BMI and attentional bias toward food and body shape to better inform treatment approaches for AN, such as the potential benefits of attention training.

### 3.4. Impulsivity

Impulsivity is a multi-faceted concept used to describe a tendency to act quickly without conscious judgement, focus, or planning and without considering the potential consequences, particularly in emotionally charged situations [39]. However, researchers in EDs have yet to agree on a standard operationalization of impulsivity [40]. Impulsivity is frequently linked to EDs presenting with binge–purge symptomatology, including the AN-BP subtype [1,41].

Studies indicate that individuals with AN-BP display higher impulsivity and greater novelty-seeking behaviours than HCs and those with the AN-R subtype [41,42]. Additionally, those with AN-BP are more likely to engage in impulsive, high-risk activities, such as substance use [42] and shoplifting [43] than HCs and those with AN-R. Impulsivity may contribute to cycles of disordered eating, like bingeing and purging, by making it difficult to resist urges, potentially worsening symptoms [1]. This subtype difference implies that AN-BP interventions may benefit from targeting impulsivity and related behaviours like substance use and binge–purge cycles in addition to other AN training methods such as managing rigidity and overcontrol.

### 3.5. Emotional Processing Difficulties

Emotional processing difficulties, such as challenges in recognizing and being aware of different emotions, have been linked to AN [25]. Research indicates that individuals with AN often struggle with identifying and managing emotions, which may lead to maladaptive behaviours, such as restrictive eating, as a coping strategy [44,45]. Emotion recognition—the ability to identify and interpret others’ emotional expressions—is essential for social functioning and mental health, and emerging evidence suggests that individuals with AN exhibit significant deficits in this area compared to HCs [46,47]. Several systematic reviews have emphasized impaired emotion recognition and altered responses to social-affective stimuli in individuals with EDs, including those with AN [48,49].

Research consistently indicates that individuals with AN also experience significant challenges in emotion regulation, manifesting as difficulties in identifying, expressing, and managing emotions—factors that contribute to the persistence of restrictive eating behaviours and heightened symptom severity [50]. Women with AN have been found to report significantly greater emotion regulation difficulties than HCs [47,50,51]. A central element of these emotional difficulties is alexithymia, defined by an impaired ability to recognize and describe one’s emotions, which is notably prevalent in AN and intensifies emotion regulation difficulties by limiting emotional awareness [52,53]. Emotion regulation challenges, coupled with alexithymia, often lead to maladaptive coping strategies such as emotional suppression and avoidance, which further reinforce disordered eating patterns and complicate treatment [53]. Consequently, addressing emotion regulation deficits, particularly by enhancing emotional awareness and fostering adaptive coping strategies, is critical for effective AN treatment.

The relationship between emotion processing (recognition and regulation) deficits and BMI in AN is inconsistent, possibly due to studies being conducted in various phases of the illness [25]. For instance, while one study found a positive correlation between BMI and emotion processing difficulties in AN such that lower BMI was associated with fewer emotion regulation difficulties [51], others have not found a significant relationship between BMI and emotion regulation [46]. A recent 15-year longitudinal study found that as individuals with AN recovered and their BMI increased, there was no significant linear effect on their emotional processing [25]. This finding is consistent with earlier studies that showed no significant association between BMI and impaired emotional functioning, whether in the acute or recovered phases of AN [45,46,54]. From such findings, Castro et al. (2021) [25] suggested that emotional dysfunction in AN may not solely be a result of malnutrition, highlighting the transdiagnostic role of emotion processing in the development and maintenance of psychiatric disorders like AN.

Finally, it is also worth outlining that poor emotion regulation strategies can strain the therapeutic relationship, as individuals with AN may struggle to express overwhelming emotions, potentially making therapy less effective or even invalidating [55]. Consequently, therapy should focus on helping clients recognize, label, and tolerate emotions while building effective coping strategies.

## 4. Bulimia Nervosa

BN is characterized by recurrent episodes of binge eating, where a person consumes an unusually large amount of food within a short period (e.g., 2 h period) and is accompanied by a sense of loss of control [4,5]. To prevent weight gain, individuals with BN typically engage in compensatory behaviours, such as self-induced vomiting, excessive exercise, or misuse of laxatives. Binge eating episodes and purging behaviours need to occur at least once a week for three months for a diagnosis of BN [4,5]. It has been suggested that the bingeing and purging behaviours reflect deficiencies in inhibitory control [56], a core executive function that overrides impulses through conscious decision-making. The exploration of neurocognition in BN has also highlighted impaired decision-making compared to those with AN [57] and HCs [58,59,60].

### 4.1. Cognitive Flexibility

Cognitive flexibility research in BN has shown mixed results. Early research suggested poor cognitive flexibility in people diagnosed with BN compared to HCs [61,62,63,64]. In contrast, another study found no significant differences in cognitive flexibility between participants with BN and HCs [65]. It also remains unknown if/how cognitive flexibility may be associated with BMI in BN. Given the limited amount of research that has investigated cognitive flexibility in BN specifically, the potential role of cognitive flexibility in treatment outcomes remains unclear.

### 4.2. Reward Processing

Reward processing refers to the neural and psychological mechanisms that evaluate, respond to, and guide behaviour based on rewarding stimuli or outcomes [66]. Such processing may be understood using a dual-pathway model [67], wherein decisions are influenced by two processes—an automatic, unconscious process primarily associated with reward sensitivity to food and a reflective, conscious process involving executive functioning that can counteract automatic processing.

In BN, a weakened reflective system is overridden by hyper-responsive automatic reward processing towards food stimuli, which manifests as an “addiction” to highly palatable foods [68]. Such heightened reward sensitivities among individuals with BN are consistently reported in the literature using self-report measures [69]. Studies have also used neurocognitive tasks such as the Game of Dice task [59] and the Iowa Gambling Task [57,58,60] to demonstrate that individuals with BN consistently prefer high-reward options despite the high risk compared to HCs. Some of these studies also showed that performance on the Iowa Gambling Task was negatively correlated with self-reported BN symptomatology [58,60]. Collectively, the evidence suggests that an impaired ability to weigh immediate rewards versus long-term consequences may fuel destructive binge–purge habits in BN. 

Another concept that may help inform the abnormalities in BN reward processing is delay discounting. It refers to a phenomenon wherein the value of a reward depreciates with a temporal delay in its delivery [70]. To our knowledge, only four papers to date have studied delay discounting in BN populations [71,72,73,74]. A meta-analysis of the three earliest papers showed an elevated discounting of monetary rewards (i.e., greater preference for sooner and smaller amounts of money) in participants with BN compared to controls, reflecting an impulsive preference for immediate rewards over delayed gratification [75]. However, the meta-analysis comprised a small, pooled sample size of 84 BN participants, and the results concerning monetary rewards may not translate to condition-specific triggers (i.e., food).

A more recent study [74] extended experimental stimuli to food rewards and observed decreased delay discounting to both monetary and food rewards in women with BN compared to HCs. That is, women with BN showed a preference for larger/later over smaller/sooner amounts of food and money relative to HCs. Although this finding contradicts previous findings from the earlier meta-analysis [75], decreased delay discounting may explain prolonged periods of food intake restriction in BN, a behaviour that necessitates a high degree of cognitive control.

### 4.3. Impulsivity

Impulsivity is implicated in BN, but findings from behavioural tasks are mixed. For example, using the Go/No-Go task, which is proposed to measure action inhibition and restraint, Rosval et al. (2006) [76] found that BN individuals did not differ significantly in overall performance compared to HCs and individuals diagnosed with AN, while Van den Eynde et al. (2012) [77] found that BN individuals showed no significant difference in commission errors when compared to HCs only. Another task, the Stroop Test, which evaluates interference/inhibitory control, was used in two studies, one of which found that BN individuals performed comparably to HCs [77] while the other study found that BN individuals performed significantly poorer than HCs [78]. Overall, the evolving body of evidence has prompted a critical re-evaluation of BN as simply a disorder of impulsivity (for more details, see the systematic review by Howard et al. (2020) [1]).

In recognition of the multi-faceted nature of impulsivity and the general nature of the aforementioned behavioural tasks, some studies have used symptom-specific assessments to study BN behaviour. For example, one study adapted the Stroop Test to include craving-specific food images as interference stimuli and found that BN individuals performed with lower accuracy than HCs [79]. Another study [80] revealed that women with BN used both health and taste ratings in making uncontrolled food choices, whereas HCs only used tastiness in their food choices. This finding suggests that binge-related food choices in BN are more calculated and less impulsive than previously thought.

In another study [71] using a computerized version of the Race Game modified to include food images designed to provoke cravings, BN individuals also demonstrated superior planning ability based on backward reasoning compared to HCs, which challenges the simplistic view of BN as a purely impulsive disorder. The authors postulated that planning, which was induced in this experiment, may work to reduce cravings for binge foods and extend binge refractory periods. This finding is especially relevant since some BN individuals deliberately plan for binge episodes in advance, and binge anticipation may decrease the motivation to attempt other coping strategies, resulting in negative reinforcement eating expectancies (i.e., the belief that eating will help mitigate distress) [81].

Thus, therapeutic strategies for BN may benefit from considering the duality of impulsivity co-existing alongside possible goal-oriented binge planning. This nuanced understanding will help inform interventions that add to existing urge surfing strategies, a technique that forms a part of cognitive behaviour therapy (CBT) for EDs and advocates “riding out” impulses by observing them instead of acting on them [82]. By identifying triggers that precede impulses and binge planning, these strategies can better target “at-risk” periods of the binge–purge cycle and inform alternative coping mechanisms that weaken negative reinforcement eating expectancies [83].

### 4.4. Emotional Processing Difficulties

Emotional processing difficulties have been revealed in individuals with BN. Some research has shown that participants with BN self-report significantly more emotion regulation difficulties than HCs, but do not exhibit difficulties in recognizing emotions [46]. A recent meta-analysis by Mourilhe et al. (2021) [84] reported a positive correlation between the amount of food ingested during a binge eating episode and depressive symptoms in BN individuals. Similarly, Davis and Smith (2023) [85] reported that positive urgency (i.e., the tendency to act impulsively in response to distress or extremely positive emotions) was associated with a greater amount of ingested food during a binge eating episode during an ad-lib meal test among women with BN. These findings may be explained by the fact that foods typically eaten during a binge eating episode are highly palatable and rewarding. Thus, the loss of control during binge episodes may be driven by the instantaneous feelings of the reward related to food consumption [86].

Additionally, individuals with BN often report heightened emotional reactivity, including difficulty managing distressing emotions, which may fuel the urge to binge eat as a form of self-soothing [87]. According to Smyth et al. (2007) [88], negative mood states such as anxiety and sadness commonly precede binge eating episodes, suggesting that binge eating may serve as an emotional escape strategy. Complementing this theory, Fischer et al. (2018) [89] found that individuals with BN are more likely to binge after experiencing negative social interactions, pointing to a connection between interpersonal stress and binge eating urges. Collectively, the evidence supports the view that BN behaviours are intertwined with a maladaptive cycle of affect regulation, where binge eating serves as a temporary escape from negative emotions.

In summary, BN is characterized by dysregulated reward processing, with heightened sensitivity to food rewards, suggesting that the immediate pleasure of binge eating plays a significant role in the loss of control. Additionally, there is some evidence to suggest that individuals with BN may struggle with emotional processing difficulties and that these difficulties drive binge eating behaviours as coping mechanisms for stress.

## 5. Binge Eating Disorder

According to the DSM-5 [4,5], BED is characterized by recurrent episodes of binge eating, accompanied by a lack of control and overeating during these episodes. Individuals with BED often eat more rapidly than usual, continue eating despite feeling uncomfortably full, eat large amounts without hunger, and may feel distress, guilt, or shame afterward [4,5]. Unlike BN, BED does not involve regular compensatory behaviours (e.g., purging) to counteract the binge episodes [4,5]. Recent research has shed light on the cognitive factors associated with BED in clinical populations [22] and non-clinical community populations [90], which may help inform our understanding of the processes involved in BED. 

### 5.1. Cognitive Flexibility

Cognitive flexibility has been investigated in a small number of studies and found to be consistently impaired in individuals with BED compared to HCs [22,91,92,93,94,95]. These investigations reveal that individuals with BED struggle to adapt their behaviour and cognitive strategies when faced with changing contingencies or rules [22,95]. Moreover, compared to patients with AN, the cognitive profile of BED has been shown to be characterized by poorer cognitive flexibility [22]. Notably, research suggests that cognitive rigidity is not merely a static trait but a dynamic process that can evolve over time. For instance, Messer et al. (2024) [96] demonstrated a significant longitudinal pathway where higher shape and weight overvaluation predicted an increased inflexible adherence to food rules, which subsequently predicted increased binge eating symptoms over a 6-month period. Complementing these findings, early childhood studies by Steegers et al. (2021) [97] revealed that set-shifting difficulties are observable even before the typical onset of EDs.

Cognitive rigidity in BED may contribute to the persistence of maladaptive eating patterns by hindering their ability to engage in adaptive problem solving [61] such as modifying established habits and routines surrounding food and eating. Particularly compelling is the “escape from awareness” theory [98], which provides a nuanced framework for understanding binge eating behaviours. This theoretical approach proposes that binge eating emerges as a complex coping mechanism, whereby individuals attempt to draw attention away from emotional distress by narrowing their cognitive focus to the immediate environment (food) rather than confronting aversive self-perceptions [99]. Consequently, individuals with BED may find it challenging to shift their focus away from food-related stimuli, resulting in increased preoccupation with food, heightened cravings, and ultimately binge eating episodes.

This perspective suggests that cognitive inflexibility is not just a symptom, but a potentially adaptive (albeit maladaptive in the long term) response to underlying emotional distress. Specific intervention strategies might include cognitive rigidity and meta-cognitive training, which helps individuals recognize and modify rigid thinking patterns [100], and acceptance-based therapies that teach individuals to experience emotions without resorting to binge eating as an avoidance mechanism [82].

### 5.2. Impulsivity

Impulsivity, particularly impaired response inhibition, has been consistently and robustly observed in individuals with BED (for reviews see [101,102,103]). Longitudinal research provides critical insights into the developmental trajectory of impulsivity in EDs. For instance, Evans et al. (2019) [104] demonstrated that higher levels of impulsivity predicted the development of disordered eating attitudes over time, suggesting impulsivity may be a dynamic risk factor rather than a static trait. Pearson et al. (2015) [105] further illuminated this relationship, revealing that negative urgency and negative affect in children can predict early binge eating engagement through distinct pathways.

The Go/No-Go Task and the Stop Signal Reaction Time Task are commonly used to assess response inhibition. Studies have found that individuals with BED exhibit increased commission errors on the Go/No-Go task and longer stop signal reaction times on the Stop Signal Reaction Time Task [101,102,103]. These errors and poor reaction times are indicative of impulsivity and poor response inhibition. An impaired ability to inhibit pre-potent responses may contribute to the difficulty in resisting the urge to binge eat, particularly in the presence of tempting or palatable stimuli. This automatic tendency to consume foods despite the individual’s awareness of the potential adverse outcomes may result in the initiation or continuation of binge eating episodes [101,102,103].

To summarize, the literature has shown that impulsivity is a complex, developmentally informed process that interacts with emotional and cognitive mechanisms to potentially maintain binge eating behaviours. Future research should continue to explore the longitudinal pathways and potential intervention points in these developmental trajectories.

### 5.3. Emotional Processing Difficulties

Emotional processing difficulties have been implicated in the onset and maintenance of BED [106], with research consistently demonstrating greater emotion-related difficulties in BED patients compared to HCs (for a review, see [3]). Longitudinal research provides crucial insights into the dynamic nature of these emotional processing difficulties, revealing that emotion regulation is not a static trait but a fluid process that interacts with ED symptoms [107].

The aforementioned “escape from awareness” theory offers a nuanced framework for understanding this relationship, proposing that binge eating serves as a cognitive mechanism to avoid confronting overwhelming emotional experiences [98,99]. Specifically, individuals with BED may use food and binge eating as a strategic method of emotional avoidance, narrowing their cognitive focus to the immediate sensory experience of eating. This process is believed to block out emotional distress, providing temporary psychological relief [98]. Negative emotions, coupled with an inability to effectively process these feelings, may trigger binge eating as a maladaptive form of emotional regulation—a means of escaping awareness of underlying psychological discomfort [98,106]. The theory explains why individuals often report feeling “numb” or “disconnected” during binge episodes, highlighting binge eating not as a simple lack of self-control, but as a complex coping mechanism for managing emotional distress. Importantly, Bodell et al.’s (2019) [107] research suggests that emotional processing difficulties in BED are dynamic, context-dependent processes that can fluctuate over time and potentially be modified through targeted, momentary interventions.

In summary, deficits in cognitive flexibility and response inhibition and emotional processing difficulties have been consistently observed in people diagnosed with BED. These cognitive impairments may contribute to the persistence of maladaptive eating patterns and the difficulty in regulating eating behaviour by hindering the ability to modify established habits, redirect attention away from food-related cues, and resist the urge to binge eat.

## 6. Otherwise Specified Feeding or Eating Disorders

OSFEDs encompass a range of disordered eating behaviours that do not fully meet the criteria for AN, BN, or BED. This category is important as it captures the experiences of many individuals who struggle with significant eating and body image disturbances but do not fit neatly into established diagnostic categories. Among the subtypes of OSFEDs, five prominent forms include: atypical anorexia nervosa (AAN), purging disorder (PD), night eating syndrome (NES), and BN and BED of low frequency and/or duration (sub-BN and sub-BED) [4,5].

Most research on OSFEDs has assessed the disorder as an overall OSFED group, treating it as a single entity despite its heterogeneous nature. This approach overlooks the possibility that distinct neurocognitive profiles may exist within OSFED’s specific subtypes. Because studies on neurocognitive experimental tasks specific to the OSFED subtypes are limited, we also include neuroimaging findings in the subsequent section, which offer valuable insights into the neural underpinnings of these cognitive processes [108,109]. Neuroimaging studies allow us to examine the activation patterns in brain regions related to reward processing, cognitive control, and emotional regulation, providing a more comprehensive understanding of the neurocognitive characteristics of OSFEDs [31].

### 6.1. Overall OSFED

The investigation of neurocognitive factors in OSFEDs remains limited, as evidenced by a review focusing on socio-cognitive factors and EDs in young people that found that only four out of the thirty-eight studies included were on OSFEDs [110]. Two studies highlighted that individuals with OSFED exhibited hyperactivation in reward, attention, and visual processing regions in response to high-calorie food cues, while showing hypo-activation in cognitive control areas, indicating difficulties in regulating responses to food-related stimuli [111,112]. However, it needs to be acknowledged that while the study by Bartholdy et al. (2019) [112] assessed a binge purging presentation, the study by Wang et al. (2016) [111] comprised a restrictive OSFED sample. Given the heterogeneous OSFED presentations in these two studies and the small number of participants, drawing clear conclusions on neurocognition in OSFEDs is challenging.

A further finding of the Prince et al. (2022) [110] review was that OSFED participants demonstrated increased neural responses in the prefrontal circuitry and cerebellum when exposed to food images, correlating with heightened obsessive–compulsive symptoms. Finally, Bodell et al. (2018) [113], also included in the Prince et al. (2022) [110] review, found that alterations in reward-related neural circuitry were concurrently and prospectively associated with binge eating in a community-based sample of adolescent girls presenting with OSFED. It should be noted that, while promising, these findings must be interpreted cautiously due to the limited sample sizes and methodological heterogeneity across neuroimaging studies, which constrain the generalizability and robustness of current conclusions.

Taken together, these findings suggest that individuals with OSFED may experience unique cognitive and emotional responses to food cues, which could impact treatment approaches and necessitate further research to develop targeted interventions.

Finally, a recent study by Wang et al. (2021) [114], not included in the Prince et al. (2022) [110] review, assessed female adolescents in a residential ED programme who had been diagnosed with AN, BN, or OSFED. Findings showed that, in comparison to a historical sample of adolescent HCs, adolescents with EDs had significantly greater levels of cognitive rigidity and attention to detail, with both findings presenting large effect sizes. Even after adjusting for age, length of illness, and BMI, there was still a substantial correlation between these neurocognitive deficits and the severity of the EDs (assessed through a clinical ED screening tool). However, in this study, the OSFED sample was very small and not separated from the other EDs. Given these limitations, the findings of Wang et al. (2021) [114] should be interpreted with caution.

#### Translation from Neuroimaging Findings to Neurocognitive Processes

The observed hyperactivation in reward, attention, and visual processing regions, coupled with hypo-activation in cognitive control areas, underscores a potential vulnerability in individuals with OSFED to food-related stimuli, potentially contributing to disordered eating behaviours. Clinically, these findings suggest that interventions targeting cognitive control processes, such as CRT [31], could help enhance regulatory responses to food cues in OSFEDs.

Similarly, the heightened prefrontal and cerebellar activity observed in response to food images, alongside obsessive–compulsive tendencies, may indicate that treatments addressing obsessive–compulsive symptoms, such as exposure and response prevention (ERP) [115], could be beneficial for certain OSFED presentations.

Finally, alterations in reward-related neural circuitry associated with binge eating symptoms in the OSFED subtypes displaying these symptoms may suggest that approaches aimed at modulating reward sensitivity—such as mindfulness-based interventions [116] or reward restructuring therapies [117]—could mitigate binge eating behaviours by helping individuals develop healthier reward mechanisms.

Future research should focus on validating these neuroimaging findings in larger, more diverse OSFED samples to refine our understanding of these neural mechanisms and their functional implications. Upcoming research should also further explore OSFED subtype-specific neurocognitive traits and compare them to the more-established profiles observed in AN, BN, and BED to enhance diagnostic precision and tailored interventions.

### 6.2. AAN

AAN is characterized by the same behavioural patterns and cognitive features as AN, but individuals diagnosed with this ED maintain a weight within or above the normal range [4,5]. Often, people with AAN are initially at a higher weight when they develop their ED. Although they may experience a dramatic weight loss, they do not drop to a BMI below 18 and are not classified as underweight [108,109].

People with AAN often experience severe cognitive distortions related to body image, including a pervasive fear of gaining weight and a preoccupation with food and dieting [118]. It is likely that cognitive rigidity and central coherence deficiencies are prevalent in this group, mirroring those observed in individuals with threshold AN. However, research that has assessed neurocognition in AAN and compared ANN to AN and/or other EDs or other comparison groups is currently lacking.

One study [119] found that in a sample of female adolescents with AAN and HCs, there was evidence for increased functional connectivity within the somatosensory network in response to food images in the AAN group. In addition, there was evidence for decreased functional connectivity in the ANN group in the brain networks linked with salience, attention, and inhibitory control and areas of the brain involved in food cue processing and appetite regulation [119]. The findings suggest that high-caloric food is associated with increased somatosensory processing in AAN, but low-caloric food is given greater salience and is considered more engaging [119]. This connectivity pattern may play a key role in the unique challenges AAN individuals face regarding food reward processing [119].

Conversely, another study by the same author found that AAN in newly diagnosed adolescents was not associated with structural changes in the brain [120]. That is, adolescents with AAN and HCs did not significantly differ in brain grey matter volume [120]. This finding contrasts with similar work in adult and adolescent AN (for a review, see [121]), often showing alterations in brain volume and structure, possibly due to factors like malnutrition and prolonged illness. Future research is required to disentangle these findings to clarify whether AAN may or may not involve similar or distinct neurobiological impacts on brain structure as seen in AN.

### 6.3. PD

PD refers to individuals who engage in purging behaviours—such as self-induced vomiting or excessive exercise—without the binge eating episodes characteristic of BN [4,5]. Negative urgency—the propensity to act impulsively when experiencing negative emotions—along with other dimensions of impulsivity, has been investigated in individuals with PD. A study by Davis et al. (2020) [122] found that individuals with PD exhibited significantly higher levels of negative urgency compared to HC, although these levels were lower than those observed in individuals with BN. However, no significant differences were detected between the PD and HC groups or between the PD and BN groups on additional impulsivity traits, including lack of premeditation, lack of perseverance, and sensation seeking [122].

Consistent with these findings, a more recent study examining a larger sample of individuals with PD reported no significant differences in sensation seeking compared to controls. Furthermore, individuals with PD were characterized by lower reward dependence and novelty seeking but exhibited higher persistence scores compared to those with BN and BED [123]. It is possible that people with PD have a tendency toward impulsive actions under emotional distress, which may reinforce the use of purging behaviours as a maladaptive coping strategy, complicating symptom management and intervention approaches for PD [122]. However, the currently limited and contradictory findings in the literature need to be clarified before such conclusions can be made.

### 6.4. NES

NES, another subtype of OSFED, involves recurrent episodes of significant food intake after the evening meal or during nighttime awakenings, often causing distress [4,5]. Though neurocognitive research is sparse, individuals with NES may have altered reward sensitivity and emotional regulation difficulties, potentially exacerbating night eating and leading to weight gain [124,125]. Like the other OSFEDs, NES also appears to be influenced by emotional fluctuations, with some individuals using night eating as a temporary coping mechanism for stress, which disrupts circadian rhythms and further heightens distress [126,127]. It is also possible that impulsivity may increase susceptibility to night binge eating episodes by weakening self-control, though this theory requires further investigation. Further research is needed to clarify the neurocognitive underpinnings of NES and its relationship with impulsivity, emotional regulation, and circadian disruptions, which could inform targeted interventions for this understudied OSFED subtype.

### 6.5. Sub-BN/BED

Sub-BN involves recurrent binge eating episodes followed by compensatory behaviours occurring less than once a week or for less than three months, while sub-BED features binge eating episodes with distress that occur less frequently than what is required for a full BED diagnosis [4,5]. In a study by Darcy et al. (2012) [128], neurocognitive assessments in adolescents with BN, sub-BN, and HCs revealed no significant group differences in set-shifting abilities, with small effect sizes. Neurocognitive factors in sub-BED have also been assessed [129,130]. For instance, Manasse et al. (2015) [129] found that individuals with sub-BED exhibited similar deficits in inhibitory control as those with full-threshold BED.

Overall, these observations suggest that neurocognitive factors may play a role in the development and maintenance of binge eating and purging behaviours along a continuum, rather than being specific to the clinical BN/BED diagnosis. These cognitive impairments may contribute to the perpetuation of maladaptive eating patterns and the potential escalation of binge eating/purging behaviours over time.

Despite the emergence of research on neurocognitive factors associated with specific OSFED subtypes, the literature remains limited compared to that of AN, BN, and BED. The heterogeneity of OSFEDs presents challenges in establishing consistent neurocognitive profiles. Factors such as varying symptomatology, comorbid conditions, and developmental influences further complicate the analysis of cognitive impairments across the different OSFED subtypes [108]. Moreover, the lack of standardized diagnostic assessment tools specifically designed for OSFEDs [109] restricts our understanding of cognitive functioning in this population. Future studies should prioritize examining cognitive and emotional processes across various OSFED subtypes to inform tailored interventions, ultimately improving treatment outcomes for this underrepresented population.

## 7. Higher Weight

Neurocognitive profiles of HW individuals highlight deficiencies in executive function, reward processing, and reward sensitivity, alongside marked changes in neuroplasticity and cognitive function in brain regions modulating reward, learning, and decision-making [10,131]. HW and neurocognitive alterations (such as impulsivity and cognitive rigidity) have been found to exhibit a bi-directional relationship; neurocognitive patterns are suggested to influence the susceptibility to overconsumption and, reciprocally, weight status and diet-elicited HW promotes changes in the prefrontal cortex structure and functionality, worsening executive functions and episodic memory [132,133,134]. For instance, in a longitudinal study spanning 9 years, HW was negatively correlated with episodic memory without any changes in executive function [135]. However, the same study found that improved executive function over time corresponded to a decline in HW. Other research has shown that executive function deficits dysregulate eating behaviours and impair physical activity functioning as risk factors for HW [136]. These findings highlight the critical role of neurocognitive processes in shaping the risk and maintenance of HW, offering potential targets for intervention.

### 7.1. Cognitive Flexibility

Cognitive flexibility has been assessed in those with HW and a meta-analysis of 25 studies found that cognitive flexibility was significantly poorer in HW participants compared to HCs [137]. Supporting this meta-analysis, a recent study also found significantly poorer flexibility in patients with HW compared to HCs [8]. Given these consistent findings, Testa et al. (2021) [8] suggested that cognitive rigidity may contribute to the persistence of unhealthy eating habits and difficulties in behaviour change. In a cognitive control task, HW participants were found to perform significantly poorer than HCs for food stimuli but not neutral stimuli (flowers) [138]. This result could indicate an impairment in cognitive flexibility, with greater cognitive resource demands in response to food stimuli [138].

### 7.2. Impulsivity

Impulsivity has been suggested to be elevated in those at a HW. A meta-analysis by Yang et al. (2018) [137] revealed that HW participants performed worse than HCs on inhibition tasks and decision-making tasks. Another systematic review by Lavagnino et al. (2016) [102] concluded that HW participants exhibit decreased inhibitory response performance in tasks when compared to HCs. Contrarily, Testa et al. (2021) [8] found greater impulsivity in participants with both HW and Type 2 Diabetes compared to HCs but no significant difference in impulsivity in those with HW compared to HCs. In a more recent study by Reyes et al. (2024) [10], BMI was correlated with task performance such that a higher BMI was associated with lower inhibitory control. The literature suggests that HW individuals tend to prioritize immediate rewards over potential future consequences, which is linked to difficulties in maintaining self-control and regulating responses to reward and punishment [8,139].

### 7.3. Reward Processing

Reward processing, the appraisal of reward and the subsequent capacity to inhibit gratification, is believed to guide individual eating behaviours in HW individuals [140]. People at an HW appear to show marked changes in reward sensitivity to food cues [141]. In a study comparing HW women to HCs, HW participants were shown to be more likely to choose immediate reward and gratification and forgo future gains, i.e., participants with an HW displayed lower delay discounting compared to HCs [9]. This low delay discounting is proposed to reflect the eating habits of HW individuals, whereby individuals may preferentially elect for immediate reward over a greater potential future benefit of improved health [9].

### 7.4. Emotional Processing Difficulties

Emotional processing difficulties have been implicated in the development of maladaptive eating behaviours contributing to HW, with a common trend of low levels of recognition, self-reporting, and emotion regulation [11,142]. In a systematic review and meta-analysis, Fernandes et al. (2018) [11] found that compared to HCs, participants at a HW had greater difficulty in identifying emotions, higher alexithymia, lower emotional awareness, and greater difficulties using emotion regulation strategies. However, there was no research to support a hypothesis that people at a HW exhibited an impaired ability to recognize the emotions of others or express their emotions [11]. Although further research is required, it appears that emotion processing in HW is nuanced and that HW is not characterized by a general emotion processing deficit [11].

Emotional factors and subsequent emotional regulation have also been found to underscore HW development. Brain regions such as the limbic system and frontal cortex, which play key roles in modulating emotions, have been revealed to contribute to the overriding of homeostatic mechanisms of feeding in HW [143]. The cortico-limbic system is believed to drive effector mechanisms in the absence of repletion signals—physiological cues indicating satiety or fullness, such as hormonal and neural signals that suppress appetite—but in the presence of potential hedonic reward signals, which promote eating for pleasure rather than need [136,144].

It should also be noted that psychological distress and emotional regulation difficulties are key risk factors for HW. Overeating often emerges as an affective response to anxiety, sadness, or stress, reflecting maladaptive coping strategies when individuals struggle to regulate emotions [145]. Impaired self-regulation and ruminative thinking can drive emotional eating, where food is used to manage negative emotions. The “escape from awareness” theory [98] suggests that individuals overeat to avoid distressing thoughts or aversive self-awareness by focusing on immediate sensory experiences, such as the taste of food. Physiologically, chronic stress and abnormal cortisol secretion can stimulate appetite, promoting cravings for high-calorie, fatty, and sweet foods by activating brain reward systems and suppressing stress response pathways [145,146]. Additionally, emotional regulation challenges may contribute to other HW risk factors, such as sedentary behaviour and sleep disturbances, further exacerbating weight gain [146].

Collectively, these findings highlight the critical interplay between psychological stress, impaired emotional regulation, physiological responses, and neurocognitive factors, such as deficiencies in cognitive flexibility, impulsivity, reward processing, and emotion regulation difficulties, driving maladaptive eating behaviours and contributing to overeating and weight gain in people at an HW.

## 8. Clinical Implications

### 8.1. Classification

Neurocognitive knowledge can significantly enhance the classification of EDs by identifying cognitive profiles that differentiate these conditions beyond the DSM-5 [4,5] or ICD-11 [28] diagnostic criteria. The variations in cognitive flexibility, central coherence, attentional biases, emotional processing, and reward processes/impulsivity outlined above across the different ED subtypes and HW (see Table 1 for an overview) can aid in refining diagnostic boundaries, offering a more granular classification system that accounts for cognitive profiles alongside behavioural symptoms (e.g., binge eating behaviour, restriction) [147]. Importantly, some of these neurocognitive factors, such as cognitive flexibility, appear to be transdiagnostic, meaning they may be present across multiple ED subtypes.

By incorporating these cognitive traits into future classification models, we can enhance the identification of distinct subtypes within and across different EDs. This approach would shift towards a dimensional framework that emphasizes underlying neurocognitive features, rather than relying solely on traditional diagnostic categories [8,75]. This model does not imply collapsing all ED diagnoses into a single group but rather focusing on the neurocognitive profiles that might cut across different ED subtypes, helping to tailor treatment based on individual cognitive characteristics.

### 8.2. Treatment

An important question in understanding neurocognitive factors in EDs is whether potential deficits in cognitive functioning represent stable traits or whether they are state factors associated with the illness. Are these deficits likely to remit on their own with recovery, or do they require targeted intervention [148]? Addressing this issue is crucial for developing effective treatment strategies. While some cognitive impairments may improve as individuals recover from the disorder, others may persist and need to be specifically targeted in treatment to optimize outcomes. Understanding the nature of these deficits—whether trait-like or state-dependent—can help clinicians decide whether to focus on cognitive remediation as part of the therapeutic process or whether to simply monitor cognitive functioning throughout recovery [148]. Presently, it remains unclear if neurocognitive impairments in EDs are traits or state factors and further research is needed to understand this issue.

Understanding the neurocognitive profiles in individuals with AN, BN, BED, OSFED, and HW is crucial for developing effective treatment interventions. In a very recent Delphi consensus paper [149], improving cognitive flexibility and impulse control were identified by researchers, clinicians, careers, and those with lived experience as key targets for ED treatments. In AN, potential deficits in cognitive flexibility, attentional biases, and emotional processing difficulties reveal specific targets for therapy. CRT (e.g., [31]) has been developed to help patients improve cognitive flexibility and central coherence, in addition to preparing patients to participate in other therapies [150].

However, a recent meta-analysis of randomized controlled trials on CRT for AN showed no significant improvement in central coherence over control treatments at the end of treatment, but this finding was based on only three studies [151]. Cognitive Remediation and Emotion Skills Training (CREST) builds on CRT by integrating cognitive training with emotion skills training. This combination has shown promising results in AN patients, with significant improvements in social anhedonia, emotional labelling, and patients’ confidence in their capacity for change [152]. Finally, it is also worth noting that there are aspects of traditional CBT that encourage adaptive and flexible thinking [153]. Additional research is therefore required to further improve CRT/CREST and develop additional treatments that can target poor cognitive flexibility across the different ED subtypes.

To address attentional biases in AN, exposure-based therapies can gradually desensitize ED individuals to anxiety-provoking food cues [115]. For instance, through repeated exposure to food-related scenarios, patients can learn to tolerate these cues without resorting to avoidance or restrictive behaviours. Furthermore, emotional processing challenges can be tackled using emotion regulation training, such as dialectical behaviour therapy (DBT) [154], which teaches ED patients to identify and manage their emotions more effectively. Skills like mindfulness [155] and distress tolerance training [156] can also enhance emotional resilience in ED individuals, potentially lowering the risk of relapse.

In BN and BED, cognitive rigidity and attentional biases may manifest in food preoccupations and impulsive behaviours. Recently, Transdiagnostic Cognitive Remediation Therapy (TCRT), a new adaptation of CRT for EDs, has been developed, which addresses common cognitive difficulties across ED diagnoses (i.e., cognitive flexibility, central coherence, and impulsivity). A recent qualitative study of thirteen patients with restrictive or binge–purge EDs and concurrent cognitive difficulties examined the impact of TCRT through semi-structured interviews [157]. Eleven participants viewed the treatment favourably, highlighting its relevance and the insights it provided into their thinking styles. Seven participants noted TCRT as a foundation for initiating changes and adopting new strategies. Engagement appeared to be enhanced by experiencing challenges directly related to their cognitive difficulties.

Other treatment modalities to consider for BN and BED, similar to AN, include attentional bias modification/training techniques, which may redirect attention away from food stimuli, helping to decrease cravings and impulsivity associated with binge episodes [35]. ERP techniques, adapted from anxiety treatments, can help BN/BED patients build a tolerance to high-craving environments [115]. Finally, adapted DPT interventions, such as DBTfor BED(DBT-BED) [158], aim to reduce binge eating by improving adaptive emotion-regulation skills and have been found to also help BED patients improve their emotion regulation difficulties.

For OSFEDs, the diverse presentations necessitate flexible and integrative interventions, as research on its classification, subtypes, and effective treatments is still in its infancy. This diagnostic category includes various symptom patterns that do not fully align with the other EDs, requiring approaches that address both the cognitive and emotional challenges specific to each presentation in addition to ED symptoms [108,109]. Combining elements from CRT/CREST, broader emotion regulation training and attentional bias modification may effectively target these unique needs. As our understanding of OSFEDs develops, refining such targeted interventions will be essential for enhancing treatment outcomes and providing tailored support.

Finally, in HW populations, impulsivity, reward processing difficulties, and emotional processing difficulties may perpetuate cycles of overeating and negatively impact body image. Mindfulness-based exercises can enhance focus and control over food-related impulses [159], while strategies from acceptance and commitment therapy and DBT [160] can help individuals reframe their relationship with food. Acceptance and commitment therapy encourages the acceptance of cravings without acting on them, fostering healthier responses to food cues and reducing binge eating behaviour [161]. Together, these interventions can foster a more adaptive relationship with food; reduce the impact of impulsivity, reward processing difficulties, and emotional biases; and enhance overall well-being.

However, it needs to be outlined that the practical implementation of these treatments requires careful consideration of their feasibility, scalability, and cultural adaptability. CRT and its adaptations (e.g., CREST and TCRT) can be delivered in both individual and group formats, increasing scalability [162]. Incorporating digital platforms or app-based interventions may further enhance accessibility, particularly for remote or underserved populations. Emotion regulation therapies, such as DBT and its adaptations (e.g., DBT-BED), have demonstrated efficacy [158], but their intensive training requirements for clinicians could pose barriers to broader implementation. Simplified versions or modular approaches might improve scalability.

Cultural adaptability is also critical. Treatments should account for variations in cultural attitudes toward food, weight, and body image, ensuring that interventions resonate with the diverse experiences of individuals across different cultural and ethnic backgrounds. Co-designing therapies with input from culturally diverse individuals with lived experience may enhance relevance and acceptability. Further research is needed to refine these approaches to ensure equitable access and culturally sensitive care for all individuals affected by EDs and HW.

#### 8.2.1. Virtual Reality

Emerging treatment modalities, such as virtual reality (VR) [163] interventions, offer innovative approaches to targeting neurocognitive deficiencies in EDs and HW populations, addressing challenges like cognitive inflexibility, attentional biases, and emotional dysregulation [163]. For attentional biases, VR can offer controlled, immersive environments where exposure-based therapies are more precisely tailored and engaging [164]. For instance, VR exposure sessions can simulate real-life food-related scenarios, allowing patients to practice exposure techniques in a safe, yet realistic setting. By engaging with these VR-based scenarios, patients could incrementally increase their tolerance for anxiety-provoking cues without using avoidance or restrictive behaviours and coping mechanisms, leading to sustained desensitization [163].

In addressing emotional processing challenges, VR can complement emotion regulation training [165] and enhance interventions like DBT [166]. By replicating challenging social or emotional contexts, VR scenarios could help individuals practice distress tolerance and mindfulness in simulated situations that they might find stressful in real life. The immersive nature of VR allows for an experiential form of learning, where patients can safely explore emotional responses, develop resilience, and refine coping skills [165,166]. This integration could lower the risk of relapse by fostering emotional resilience and enhancing self-regulation skills that are essential for recovery. In addition, given the widespread adoption of VR technologies, these skills could be practised by patients in their homes outside of medical appointments with clinicians.

#### 8.2.2. Brain Stimulation

Different brain stimulation techniques can also offer a promising approach to improving cognitive deficiencies in individuals with EDs and those with HW by directly targeting the neural circuits associated with cognitive flexibility, emotional regulation, and/or impulse control. Techniques such as transcranial magnetic stimulation [167] and transcranial direct current stimulation [168,169] can modulate neural activity in areas of the brain implicated in these cognitive domains, potentially enhancing the effectiveness of traditional therapies.

Cognitive flexibility has largely been associated with functioning in the prefrontal cortex [170], the anterior cingulate cortex [171], and the orbitofrontal cortex [172]. Stimulating these areas may increase activity in these brain regions, neural pathways, and neural plasticity, potentially improving cognitive flexibility and reducing rigid, rule-bound thinking. Similarly, individuals with BN and BED, who may exhibit heightened impulsivity and attentional biases toward food cues, could benefit from brain stimulation targeting the dorsolateral prefrontal cortex to improve inhibitory control and reduce impulsive behaviours [173,174].

In individuals with HW, brain stimulation may address impairments in reward processing and executive function, which are often associated with challenges in regulating food intake. Stimulating or inhibiting activity in the prefrontal cortex could enhance impulse control and support better decision-making in food-related contexts, potentially leading to healthier eating patterns [175,176].

## 9. Limitations

While our narrative review encompasses a broad range of studies examining neurocognitive and related factors in EDs and HW individuals, it may not be comprehensive. We recognize the necessity for additional structured systematic reviews and/or meta-analyses to further explore this topic, as these approaches would provide a more rigorous synthesis of the literature, including standardized inclusion criteria and quantitative pooling of results. The decision to conduct a narrative review, rather than a systematic review, was based on the exploratory nature of this topic and the need to provide a broad overview of diverse studies across multiple domains of neurocognitive processes. A narrative review allows for a more flexible and comprehensive discussion of varied study designs, methodologies, and findings, which may be less feasible in the structured format of a systematic review. Another limitation of this review is the exclusion of studies on ARFID and other EDsin childhood [4,5]. This decision is in line with previous reviews [89] given that these disorders typically develop in early childhood and are clinically distinct in presentation to AN, BN, BED, and OSFED [4,5].

Omitting ARFID from the review limits the comprehensiveness of understanding neurocognitive factors across the full spectrum of EDs. It overlooks unique developmental and neurocognitive challenges associated with early-onset disorders, such as sensory sensitivities and anxiety-related behaviours, which differ from those observed in other EDs [4,5]. Future research may wish to include ARFID to ensure a more comprehensive understanding of neurocognitive factors across the full spectrum of EDs, including early-onset disorders, which may present unique developmental and cognitive challenges.

There are also various limitations within the reviewed studies that need to be considered. First, the study of neurocognitive profiles in EDs and HW populations is constrained by several methodological limitations. A significant challenge lies in the diversity of cognitive assessment tools utilized across studies. This variability in tasks designed to evaluate key domains such as cognitive flexibility, attentional bias, and emotion regulation complicates cross-study comparisons and hinders the identification of shared or disorder-specific impairments. For instance, while some studies use standardized neurocognitive tests, others employ experimental paradigms or self-report measures, each with varying levels of reliability and ecological validity.

Second, many studies lack uniformity in the operational definitions and measurement of neurocognitive constructs, leading to inconsistent results. Small and non-representative sample sizes are another limitation, often reducing the generalizability of findings to broader clinical populations. Few studies stratify their samples using important demographic or clinical factors, such as age, gender, comorbidities, or the duration of the illness/condition, which may influence neurocognitive functioning.

Third, theoretical limitations also warrant consideration. Many studies fail to clearly differentiate between trait-like neurocognitive deficits that may predispose individuals to EDs and HW conditions and state-like impairments that emerge as consequences of these disorders or associated factors such as malnutrition or weight changes. This lack of theoretical clarity complicates the interpretation of findings and the development of targeted interventions. Furthermore, it remains unclear which cognitive factors are inherent traits, which exist premorbidly, which are scar effects, and which are symptoms of the ED illness and HW, underscoring the need for more nuanced research in this area.

Fourth, the reliance on cross-sectional studies restricts the understanding of how neurocognitive deficits develop and progress across different stages of ED, recovery, and in people with an HW. The limited amount of longitudinal research available on EDs and cognition restricts insights into whether neurocognitive impairments endure over time or improve with treatment and recovery. For example, longitudinal tracking of cognitive flexibility in AN could help clarify whether such deficits improve with weight restoration and recovery or if they persist as chronic features of the disorder. Similarly, observing attentional biases in BN and BED across treatment phases could reveal whether these biases diminish as part of recovery or if they remain stable, potentially contributing to relapse risk.

Fifth, it is important to acknowledge the influence of weight and nutritional status on neurocognitive function and highlight that BMI alone is insufficient to capture the full scope of malnutrition and starvation and the impact that they might have on cognition. For instance, cognitive flexibility difficulties have been found in AN, yet BMI does not account for this complexity in understanding the neurocognitive profile of AN (for a review, see [20]).

Sixth, the variability in ED subtypes and illness stages among participants, which may affect neurocognitive findings across disorders, complicates the comparison of studies. AN-R and AN-BP present with different characteristics [4,5], yet most studies have lumped the two groups into one overall AN group due to small sample sizes. The transdiagnostic nature of ED symptoms further blurs the lines between disorders, as features like impulsivity and reward sensitivity, commonly observed in AN-BP, BN and BED, may represent transdiagnostic vulnerabilities rather than disorder-specific traits. Additionally, there is likely substantial heterogeneity in cognitive deficits across EDs and within cognitive domains—some deficits may be more prominent in one ED subtype than another, reflecting variability in the nature and intensity of these impairments. 

Finally, as outlined in the current review, OSFEDs are underrepresented in the literature, and this lack of data makes it difficult to determine whether their neurocognitive profile aligns with other EDs or HW individuals.

## 10. Future Research Directions

Future research should focus on refining our understanding of neurocognitive profiles in AN, BN, BED, OSFED, and HW through detailed longitudinal studies. Longitudinal studies are needed to track cognitive changes over time and capture the dynamic nature of these disorders and their recovery phases. However, the challenges of such research need to be addressed. The key challenges include maintaining participant engagement over extended periods, managing attrition rates, and the need for sophisticated methodologies to assess cognitive changes reliably. Specific testable hypotheses for future studies should include the following question: are the observed neurocognitive deficits in these disorders temporarily associated with the current illness stage or severity, or are they stable traits that persist beyond weight restoration? It would also be important to further assess how specific neurocognitive traits predict recovery trajectories or relapse risks in EDs and HW. Future studies could, for instance, assess the following hypothesis: “Do deficits in cognitive flexibility and impulse control at the start of treatment predict poorer recovery trajectories in AN patients?” Such concrete research questions will help guide future studies in identifying predictors of treatment success and relapse, thus informing more personalized and effective intervention strategies.

Incorporating neuroimaging data can further refine this understanding. Neuroimaging studies, such as functional magnetic resonance imaging (fMRI), can elucidate changes in brain activity related to cognitive flexibility, reward processing, and other executive functions in these populations. Integrating neuroimaging with biomarkers could lead to a more comprehensive understanding of the biological underpinnings of cognitive impairments in these populations and help in designing targeted interventions that address both nutritional and cognitive deficits effectively.

Furthermore, the inclusion of different OSFED presentations and ARFID in future research is essential for comprehensively assessing the full range of cognitive impairments and their connection to emotional and behavioural patterns in individuals with disordered eating. By expanding the research focus to encompass OSFEDs, we can better understand how individuals with these illnesses navigate their cognitive challenges and develop targeted interventions that address their unique needs.

Finally, there is a need for further interdisciplinary collaborations to investigate neurocognitive profiles in EDs and HW populations. These collaborations will enable the integration of diverse expertise, including psychology, neuroscience, nutrition, and clinical treatment strategies, to develop a holistic understanding of the complex interplay between neurocognitive functioning, physiological markers, and behavioural data. Such an approach will be essential in advancing personalized treatment approaches that are responsive to the unique needs of individuals with EDs and HW.

## 11. Conclusions

To conclude, the unique contributions of the current review include its wide coverage of the most up-to-date literature on neurocognitive factors across EDs and HW, providing a comprehensive synthesis of the existing research in this area. It highlights the significance of poor cognitive flexibility and attentional biases in AN, underscoring the need for tailored interventions to address these specific cognitive patterns, such as CRT/CREST and CBT strategies. For BN and BED, this review discusses how poor cognitive flexibility, emotional processing difficulties, and heightened reward sensitivity contribute to binge eating and purging behaviours, suggesting the need for therapeutic approaches that manage these complex relationships, such as DBT-BED. Finally, individuals with HW face neurocognitive challenges like impaired decision-making and executive function, further emphasizing the need for individualized, transdiagnostic treatment strategies. Given that research on OSFED subtypes is still in its infancy, conclusions regarding OSFEDs remain preliminary and highlight the need for further investigation to better understand these subtypes and their unique cognitive challenges. Future research should prioritize longitudinal studies to explore the evolution of these neurocognitive profiles over time, conduct further neuroimaging, consider biomarkers, and collaborate with other fields. Such research will ensure the development of effective, tailored interventions that improve recovery outcomes across diverse ED and HW populations.

## Figures and Tables

**Table 1 nutrients-16-04418-t001:** Neurocognitive processes across eating disorders and higher weight.

Neurocognitive Process	Anorexia * Nervosa (AN)	Bulimia Nervosa (BN)	Binge Eating Disorder (BED)	Higher Weight (HW)
**Cognitive Flexibility**	↓ (consistently impaired; rigid food rules)	↓ (impaired; cognitive rigidity evident in binge–purge cycles)—evidence lacking	↓ (impaired, particularly in food contexts)	↓ (mild impairment, especially for food-related stimuli)
**Impulsivity**	↑ (increased inhibitory control during restriction; difficulty modulating impulsivity in binge–purge episodes)	↑ (elevated impulsivity, exacerbated by negative affect and emotional triggers)	↑ (high impulsivity; difficulty resisting binge urges; impaired response inhibition for palatable stimuli)	↑ (low inhibitory control, particularly for food stimuli; associated with BMI and immediate gratification)
**Emotional Processing**	↓ (greater difficulty identifying and processing emotions; alexithymia noted)	↓ (emotional dysregulation; binge–purge cycles as avoidance)	↓ (heightened emotional dysregulation; bingeing used as emotional avoidance)	↔ or ↓ (context-dependent; alexithymia and emotional challenges present in some individuals)
**Reward Processing**	↓ (low reward sensitivity and rigid adherence to rules)	↑ (heightened sensitivity to food-related rewards)	↑ (preference for immediate food rewards over delayed benefits)	↑ (greater preference for immediate gratification at the expense of long-term outcomes)

↑: increased/elevated; ↓: decreased; ↔: mixed or context-dependent. * Results might differ between AN-Restrictive and AN-Binge Purging Subtype. OSFEDs have been omitted from the table due to lacking research.

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
