# Peer review of "A Narrative Review on the Neurocognitive Profiles in Eating Disorders and Higher Weight Individuals: Insights for Targeted Interventions"

_nutrients, 2024, doi:10.3390/nu16244418_

Round 1
Reviewer 1 Report
Comments and Suggestions for Authors
This narrative review is an in-depth analysis of the state of play of the different DEs. However, despite the quality of the narrative, the depth of content and the accuracy of its conclusions, there are a couple of aspects that, in my view, can be improved and must be addressed.
Firstly, the fact that it is a narrative review does not exempt the authors from explaining their scientific process, which is almost completely absent. Where was the information sought? What criteria were followed to select the most relevant current scientific contributions? What protocol was followed? Were there discards? Why? In conclusion, the authors should create a Method section in which they clarify to the reader the scientific process followed.
Secondly, the poor citation method should be highlighted. APA and MDPI citation style is mixed throughout the text. Sometimes the MDPI style is missing, sometimes it is combined with APA, sometimes without a year, sometimes with a year. Authors should thoroughly review this issue, opting for the MDPI style and checking that the bibliography includes all the citations highlighted throughout the text.
Author Response
Comment 1: Firstly, the fact that it is a narrative review does not exempt the authors from explaining their scientific process, which is almost completely absent. Where was the information sought? What criteria were followed to select the most relevant current scientific contributions? What protocol was followed? Were there discards? Why? In conclusion, the authors should create a Method section in which they clarify to the reader the scientific process followed.
Response 1: Thank you for raising this. We have now added a method section to explain the scientific process how the studies included in the review were selected. Please refer to the new 2. Method section that we have added in these revisions.
Comment 2: Secondly, the poor citation method should be highlighted. APA and MDPI citation style is mixed throughout the text. Sometimes the MDPI style is missing, sometimes it is combined with APA, sometimes without a year, sometimes with a year. Authors should thoroughly review this issue, opting for the MDPI style and checking that the bibliography includes all the citations highlighted throughout the text.
Response 2: We apologise for this oversight and have now made sure all references are correctly noted using MDP style.
Reviewer 2 Report
Comments and Suggestions for Authors
Abstract
- The abstract provides an overview but lacks specificity about individual findings across eating disorders (EDs). The authors should briefly summarise findings per disorder (e.g., AN: impaired cognitive flexibility; BN: impulsivity and reward sensitivity, etc.).
- Terminologies like “neurocognitive impairments" are broad and could mislead readers regarding their focus on specific cognitive domains.
- I suggest adding explicit statements on the methodological scope (e.g., experimental vs. self-reported data).
- The authors should include a more precise articulation of clinical and research implications.
Introduction
- The authors discuss neurocognitive characteristics but are repetitive when addressing their intrinsic versus weight-related origins.
- The rationale for including HW populations is well-founded but needs a sufficient critique of potential biases in how HW is categorized in research.
- The authors should consolidate the discussion on intrinsic versus secondary neurocognitive deficits into a single paragraph.
- I’m missing critical reflection on how cultural and medical biases in HW terminology might affect findings and interpretations.
- Please expand the justification for excluding disorders like ARFID while including HW, addressing implications for the scope of the review.
Anorexia nervosa
- Subsections (e.g., cognitive flexibility, attentional bias) present findings but must coherently synthesize the clinical and theoretical implications.
- Contradictory findings on cognitive flexibility and BMI need to be reconciled effectively.
- I suggest a summary in each subsection connecting findings to treatment or theoretical models.
- The authors should discuss methodological inconsistencies, such as variations in task design or sample characteristics, that contribute to conflicting results.
- Despite specific deficits, the authors should address the implications of preserved general intelligence in AN.
Bulimia nervosa
- Reward processing findings must be more consistent and sufficiently explored to reconcile differing conclusions about heightened sensitivity versus hypo-functioning.
- The relationship between impulsivity and deliberate planning in BN needs to be developed and presented ambiguously.
- The paper would benefit from analyzing whether these contradictions stem from methodological issues, population differences, or divergent conceptualizations of reward.
- The authors should highlight the clinical significance of impulsivity coexisting with goal-oriented planning and how this duality could inform tailored interventions.
- Using specific examples to demonstrate how neurocognitive traits manifest in real-life BN behaviours would improve the author’s rationale.
Binge eating disorders
- Cognitive rigidity and emotional processing are well-covered, but the connection between these impairments and binge behaviours remains superficial.
- Limited exploration of longitudinal data on neurocognitive profiles restricts insights into whether these deficits are trait- or state-dependent.
- I suggest emphasizing mechanisms like the “escape from awareness" theory and its potential to bridge cognitive rigidity and emotional distress with binge behaviours.
- Including longitudinal findings or proposing hypotheses about the stability of neurocognitive impairments would add depth to the MS.
- Highlighting implications for therapeutic interventions targeting these cognitive-emotional links is needed.
OSFED
- The section fails to address the unique neurocognitive profiles of subtypes like atypical anorexia nervosa (AAN) and purging disorder (PD) thoroughly.
- Neuroimaging findings are described but do not discuss how they translate to functional or clinical outcomes.
- The authors should expand on subtype-specific neurocognitive traits and how they compare to established EDs.
- Critically appraise the limited sample sizes and methodological heterogeneity in neuroimaging studies.
- The authors should discuss practical implications for developing diagnostic tools and tailored treatments based on these findings.
Higher weight
- The section overemphasizes cognitive flexibility and impulsivity while neglecting broader neurocognitive and emotional factors.
- The bidirectional relationship between HW and neurocognition is mentioned but needs to be adequately explored.
- It should be discussed how external factors, such as stigma and psychological stress, might mediate the relationship between HW and neurocognitive deficits.
- Please provide a deeper examination of the interplay between HW-specific emotional regulation issues and decision-making impairments.
- Mentioning interventions that consider physiological and psychosocial factors influencing HW neurocognition must be included.
Clinical implications
- The discussion is weighted heavily towards traditional interventions like CRT and emotion regulation training, with insufficient focus on emerging therapies.
- The section needs to consider the proposed treatments' feasibility, scalability, and cultural adaptability.
- Critique the effectiveness of existing interventions like CRT and propose refinements based on the review findings.
- Recommendation of integrated neurocognitive training into current treatment paradigms for underrepresented populations (e.g., HW, OSFED) will benefit the MS depth.
Limitations
- The limitations section recognizes critical gaps but needs to include a more profound critique of the studies' methodological and theoretical shortcomings.
- The reliance on cross-sectional data is mentioned but needs to be elaborated on sufficiently to highlight its impact on findings.
- The authors should discuss how the diversity in assessment tools (e.g., self-report vs. experimental) and small sample sizes limit generalizability.
- Exploring the implications of omitting ARFID and childhood-onset EDs for the review’s conclusions is required.
- The authors can call for greater methodological rigour in future studies, including harmonizing cognitive task designs and diagnostic criteria.
Future directions
- Recommendations for future research are general and lack specific testable hypotheses or methodological frameworks.
- The emphasis on longitudinal studies is valid, but the challenges of such research need to be addressed.
- Proposing concrete research questions, such as how specific neurocognitive traits predict recovery trajectories or relapse risks, can be more relevant to readers.
- The authors can discuss practical methodologies, such as combining neuroimaging with behavioural data or integrating biomarkers.
- Addressing the need for interdisciplinary collaborations to investigate the neurobiological underpinnings of EDs and HW populations will enhance the rigidity of this sub-section.
Conclusions
- The conclusion reiterates findings without providing actionable insights or a strong narrative linking all sections.
- The need for individualized treatment approaches is emphasized but not grounded in specific examples from the review.
- A compelling argument for integrating neurocognitive findings into ED classification systems and clinical frameworks is needed.
- Emphasizing the review’s unique contributions while acknowledging its limitations should be introduced more explicitly.
General Criticisms
- Many sections repeat similar points (e.g., BMI’s role in neurocognitive deficits), which could be consolidated.
- Findings across disorders are presented in isolation, missing an opportunity to identify transdiagnostic patterns.
- OSFED and HW populations are discussed superficially compared to AN and BN, reflecting a bias in coverage.
- While the paper is well-referenced, some sections rely on excessive citations without critically evaluating sources.
Author Response
Abstract
Comment 3: The abstract provides an overview but lacks specificity about individual findings across eating disorders (EDs). The authors should briefly summarise findings per disorder (e.g., AN: impaired cognitive flexibility; BN: impulsivity and reward sensitivity, etc.).
Response 3: Thank you for this comment. We have highlighted the transdiagnostic factors and tried to make the abstract more specific in relation to the findings for the different ED subtypes and HW individuals.
The new results section of the abstract now reads as: “Results: We found that some neurocognitive characteristics, such as cognitive rigidity, impulsivity, emotion processing difficulties, and dysregulated reward processing, appear transdiagnostic, spanning multiple ED subtypes and HW populations. We also revealed neurocognitive features specific to ED subtypes and HW. For instance, individuals with AN demonstrate an enhanced focus on detail, and BN and BED are characterized by a pronounced attentional bias toward food-related stimuli. In individuals with HW cognitive processes underpin behaviours associated with overeating and weight gain.”
Comment 4: Terminologies like “neurocognitive impairments" are broad and could mislead readers regarding their focus on specific cognitive domains.
Response 4: Thank you we have now replaced the terminology “neurocognitive impairment” with “neurocognitive profiles/characteristics/features” throughout the abstract. Where necessary we also outline the specific neurocognitive domain.
Comment 5: I suggest adding explicit statements on the methodological scope (e.g., experimental vs. self-reported data).
Response 5: Thank you for this. We have now added the following sentence to the method section to make the methodological scope of the review clearer:” This critical narrative review focuses on neurocognitive findings derived from mainly experimental tasks to provide a detailed understanding of cognitive functioning across these groups. Where experimental data are lacking, we draw on self-report measures and neuroimaging findings to offer supplementary insights.
Method: A search of major databases that prioritized meta-analyses and recent publications (last 10 years) was conducted. Using comprehensive search terms related to EDs, HW, and neurocognition, eligible studies focused on human neurocognitive outcomes (e.g., cognitive flexibility, attentional bias etc.) and were published in English.”
Comment 6: The authors should include a more precise articulation of clinical and research implications.
Response 6: Thank you for this suggestion. We have now made the clinical and research section for the conclusion more specific.
We have added the following information:” Conclusions: These findings highlight the critical importance of understanding both the unique and shared neurocognitive patterns across ED subtypes and HW populations. By identifying transdiagnostic factors, such as cognitive rigidity and reward processing, alongside ED subtype/HW-specific vulnerabilities, researchers and clinicians can develop more nuanced, evidence-based interventions that address the core mechanisms driving disordered eating behaviours.”
Introduction
Comment 7: The authors discuss neurocognitive characteristics but are repetitive when addressing their intrinsic versus weight-related origins.
Response 7: We reviewed the introduction and did not identify significant repetition. Two sentences addressed the intrinsic versus weight-related origins of neurocognitive patterns, with the second emphasizing the review's importance. To address the feedback, we revised this section to reduce redundancy and better align with the opening paragraph.
Comment 8: The rationale for including HW populations is well-founded but needs a sufficient critique of potential biases in how HW is categorized in research.
Response 8: Thank you for this suggestion. We have now added the following paragraph in the introduction to make the reader aware of these biases early in the review.
“Furthermore, it is important to acknowledge potential biases in how HW is categorized in research. BMI, while widely used, does not account for body composition, distribution of fat, or metabolic health, potentially oversimplifying a complex construct [12,13]. Cultural and medical biases in the HW terminology, such as framing HW primarily as a pathological state, can also shape the focus of research and influence the interpretation of findings [13,14]. These biases may perpetuate stigmatizing narratives, overlook protective or neutral factors, and obscure the heterogeneity of HW populations. A critical examination of these issues is essential to accurately interpret findings and ensure that the inclusion of HW populations in ED research advances understanding without reinforcing stereotypes or over-simplifications.”
Comment 9: The authors should consolidate the discussion on intrinsic versus secondary neurocognitive deficits into a single paragraph.
Response 9: Thank you for this comment. We have now consolidated this information.
Comment 10: I’m missing critical reflection on how cultural and medical biases in HW terminology might affect findings and interpretations.
Response 10: Thanks for this suggestion. Please refer to Response 8 to see how we have now incorporated on how cultural and medical biases in HW terminology might affect findings and interpretations.
Comment 11: Please expand the justification for excluding disorders like ARFID while including HW, addressing implications for the scope of the review.
Response 11: We have now added a paragraph at the end of the introduction that expands the justification provided in the discussion as regards to why ARFID was not included in the review.
This new paragraph now reads as: “Avoidant/Restrictive Food Intake Disorder (ARFID) was not included in this review because it differs from traditional EDs like AN, BN, BED, OSFED, and HW in terms of its clinical presentation. ARFID is characterized by food avoidance based on sensory characteristics rather than body image concerns or binge-purge behaviours [4,5]. Therefore, its exclusion allows for a more focused examination of the neurocognitive characteristics of ED subtypes and HW populations who present with a more similar clinical presentation.”
Anorexia nervosa
Comment 12: Subsections (e.g., cognitive flexibility, attentional bias) present findings but must coherently synthesize the clinical and theoretical implications.
Response 12: Thanks for the reviewer suggestion, we have added more implications for these sections and it is read as:
Cognitive flexibility: “These findings suggest that cognitive inflexibility in AN may not be merely a consequence of low weight or starvation but may instead represent a core feature of the disorder.
Understanding cognitive inflexibility as a central aspect of AN points to promising clinical interventions. Targeting this rigidity through cognitive training such as Cognitive Remediation Therapy (CRT) [31], increasing awareness of inflexible thought patterns, and designing behavioural exercises to improve set-shifting abilities may help address key maintenance factors and support long-term recovery.
Central coherence: “In summary, although more research is needed to clarify the relationship between BMI and central coherence in AN, current evidence points to central coherence deficits among those with AN. A deeper understanding of neuropsychological functioning in AN, particularly its link to neural mechanisms and symptoms, could inform more effective treatments.”
Attention bias: “Future studies should further investigate the relationship between BMI and attentional bias toward food and body shape to better inform treatment approaches for AN, such as the potential benefits of attention training.”
Impulsivity: “This subtype difference implies that AN-BP interventions may benefit from targeting impulsivity and related behaviours like substance use and binge-purge cycles in addition to other AN training such as managing rigidity and overcontrol.”
Emotional processing difficulties: “From such findings, Castro et al. (2021) [25] suggested than emotional dysfunction in AN may not solely be a result of malnutrition, highlighting the transdiagnostic role of emotion processing in the development and maintenance of psychiatric disorders like AN. Finally, it is also worth outlining that poor emotion regulation strategies can strain the therapeutic relationship, as individuals with AN may struggle to express overwhelming emotions, potentially making therapy less effective or even invalidating [55]. Consequently, therapy should focus on helping clients recognise, label, and tolerate emotions while building effective coping strategies.”
Comment 13: Contradictory findings on cognitive flexibility and BMI need to be reconciled effectively.
Response 13: We have edited this section to make it more cohesive.
“Other research has explored the relationship between cognitive flexibility and BMI in AN as BMI is suggested as a severity indicator for AN per both the DSM-5 [4,5] and ICD-11 [28]. A recent systematic review found no association between cognitive flexibility and BMI [20], and this finding was supported by subsequent studies [29,30]. These findings suggest that cognitive inflexibility in AN may not be merely a consequence of low weight or starvation but may instead represent a core feature of the disorder.”
Comment 14: I suggest a summary in each subsection connecting findings to treatment or theoretical models.
Response 14: Thank you for raising this. We have addressed this in comment 12.
Comment 15: The authors should discuss methodological inconsistencies, such as variations in task design or sample characteristics, that contribute to conflicting results.
Response 15: Thanks for your comments. We have addressed these limitations in the AN section.
Cognitive flexibility: “Miles et al. (2020) [2] noted that the 69 studies included in their review used over 30 different measures of cognitive flexibility, creating inconsistencies that hinder synthesis and may contribute to conflicting findings. The review further notes variability in findings for weight-restored and fully recovered AN samples, suggesting that while some aspects of cognitive flexibility may improve, certain deficits may persist even after recovery or weight restoration [2].”
Central coherence:” While most findings suggest deficits in central coherence among those with AN, the findings remain somewhat inconsistent. This variability is partly due to the heterogeneity of studies and methodological differences that need to be considered. Various tests have been used to measure central coherence, raising questions about the comparability of results [2]. Different cognitive tests vary in complexity and often target multiple cognitive functions. For instance, the Homograph Reading Task focuses on linguistic and contextual coherence, whereas the Group Embedded Figures Test assesses perceptual and visual-spatial aspects, each providing distinct insights into different components of central coherence.”
Attentional bias: “Research investigating attentional processes in AN employs diverse methods, including variations in measurement procedures, types of stimuli (e.g., words or images), threat content, and outcome measures. Despite this methodological diversity, findings consistently show heightened attentional bias toward body shape, weight, and food-related stimuli in individuals with AN compared to HCs [12,35].”
Comment 16: Despite specific deficits, the authors should address the implications of preserved general intelligence in AN
Response 16: We have added the following information to address implications of preserved intelligence in AN: “While the role of starvation in AN remains unclear, certain neurocognitive functions appear preserved, with general intelligence typically within normal ranges [20]. This preservation supports treatment engagement and recovery, underscoring the importance of addressing specific deficits while leveraging premorbid cognitive strengths.”
Bulimia nervosa
Comment 17: Reward processing findings must be more consistent and sufficiently explored to reconcile differing conclusions about heightened sensitivity versus hypo-functioning.
Response 17: Thank you for this comment. This section has now been rewritten to reflect studies using neurocognitive tasks as opposed to neuroimaging (which was the previous source of contradiction regarding heightened sensitivity towards rewards in some studies vs hypo-functioning of reward circuitry in others) as neuroimaging is beyond the scope of this review (apart from the OSFED section).
This section now reads as: “Such processing may be understood based on a dual-pathway model [67], wherein decisions are influenced by two processes – an automatic, unconscious process primarily associated with reward sensitivity to food, and a reflective, conscious process involving executive functioning that can counteract automatic processing.
In BN, a weakened reflective system is over-ridden by hyper-responsive automatic reward processing towards food stimuli, which manifests as an ‘addiction’ to highly palatable foods [68]. Such heightened reward sensitivities among individuals with BN is consistently reported in the literature using self-report measures [69]. Studies have also used neurocognitive tasks such as the Game of Dice task [59] and Iowa Gambling Task [57,58,60] to demonstrate that individuals with BN consistently prefer high reward options despite high risk as compared to HCs. Some of these studies also showed that performance on the Iowa Gambling Task was negatively correlated with self-reported BN symptomatology [58,60]. Collectively, the evidence suggests that an impaired ability to weigh immediate rewards versus long-term consequences may fuel destructive binge-purge habits in BN.”
Comment 18: The relationship between impulsivity and deliberate planning in BN needs to be developed and presented ambiguously.
Response 18: This section has now also been rewritten to reflect the multi-faceted nature of impulsivity, with reference to studies that assess impulsivity in BN with general or more symptom-specific tasks, as well as deliberate planning hypotheses as explored in studies
This section now reads as: “4.3. Impulsivity is implicated in BN, but findings from behavioural tasks are mixed. For example, using the Go No-Go task, which is proposed to measure action inhibition and restraint, Rosval et al. (2006) [76]found that BN individuals did not differ significantly on overall performance compared to HCs and individuals diagnosed with AN, while Van den Eynde et al. (2012) [77] found that BN individuals showed no significant difference in commission errors when compared to HCs only. Another task, the Stroop test, which evaluates interference/inhibitory control, was used in two studies, one of which found that BN individuals performed comparable to HCs [77] while the other study found that BN individuals performed significantly poorer than HCs [78]. Overall, the evolving body of evidence has prompted critical re-evaluation of BN as simply a disorder of impulsivity (for more details, see the systematic review by Howard et al. (2020) [1]).
In recognition of the multi-faceted nature of impulsivity and the general nature of the aforementioned behavioural tasks, some studies have used symptom-specific assessments to study BN behaviour. For example, one study adapted the Stroop test to include craving-specific food images as interference stimuli and found that BN individuals performed with lower accuracy than HCs [79]. Another study [80] revealed that women with BN used both health and taste ratings in making uncontrolled food choices, whereas HCs only used tastiness in their food choices. This finding suggests that binge-related food choices in BN are more calculated and less impulsive than previously thought. In another study [71]using a computerized version of the Race Game modified to include food images designed to provoke cravings, BN individuals also demonstrated superior planning ability based on backward reasoning compared to HCs, which challenges the simplistic view of BN as a purely impulsive disorder. The authors postulated that planning, such as that which was induced in this experiment, may work to reduce cravings for binge foods and extend binge refractory periods. This finding is especially relevant since some BN individuals deliberately plan for binge episodes in advance, and binge anticipation may decrease the motivation to attempt other coping strategies, resulting in negative reinforcement eating expectancies (i.e., the belief that eating will help mitigate distress) [81].
Thus, therapeutic strategies for BN may benefit from considering the duality of impulsivity co-existing alongside possible goal-oriented binge planning. This nuanced understanding will help inform interventions that add to existing urge surfing strategies, a technique that forms a part of cognitive behaviour therapy (CBT) for EDs and advocates ‘riding out’ impulses by observing them instead of acting on them [82]. By identifying triggers that precede impulses and binge planning, these strategies can better target ‘at-risk’ periods of the binge-purge cycle and inform alternative coping mechanisms that weaken negative reinforcement eating expectancies [83].”
Comment 19: The paper would benefit from analysing whether these contradictions stem from methodological issues, population differences, or divergent conceptualizations of reward.
Response 19: Please refer to response provided to Comment 17 above.
Comment 20: The authors should highlight the clinical significance of impulsivity coexisting with goal-oriented planning and how this duality could inform tailored interventions.
Response 20: Thank you for this. We have now made the following revisions in the text: “Thus, therapeutic strategies for BN may benefit from considering the duality of impulsivity co-existing alongside possible goal-oriented binge planning. This nuanced understanding will help inform interventions that add to existing urge surfing strategies, a technique that forms a part of cognitive behaviour therapy (CBT) for EDs and advocates ‘riding out’ impulses by observing them instead of acting on them [82]. By identifying triggers that precede impulses and binge planning, these strategies can better target ‘at-risk’ periods of the binge-purge cycle and inform alternative coping mechanisms that weaken negative reinforcement eating expectancies [83].”
Comment 21: Using specific examples to demonstrate how neurocognitive traits manifest in real-life BN behaviours would improve the author’s rationale.
Response 21: Thanks for the suggestion. Examples were added in the rewritten section for reward processing: “In BN, a weakened reflective system is over-ridden by hyper-responsive automatic reward processing towards food stimuli, which manifests as an ‘addiction’ to highly palatable foods [68]. Such heightened reward sensitivities among individuals with BN is consistently reported in the literature using self-report measures [69].”
In the impulsivity section, findings regarding planning based on backward reasoning from the Race Game was further outlined that “The authors postulated that planning, such as that which was induced in this experiment, may work to reduce cravings for binge foods and extend binge refractory periods. This finding is especially relevant since some BN individuals deliberately plan for binge episodes in advance, and binge anticipation may decrease the motivation to attempt other coping strategies, resulting in negative reinforcement eating expectancies (i.e., the belief that eating will help mitigate distress) [81].”
Binge eating disorders
Comment 22: Cognitive rigidity and emotional processing are well-covered, but the connection between these impairments and binge behaviours remains superficial.
Response 22: We have addressed this in response to comment 24 (see below).
Comment 23: Limited exploration of longitudinal data on neurocognitive profiles restricts insights into whether these deficits are trait- or state-dependent.
Response 23: We have addressed this in response to comment 25 (see below).
Comment 24: I suggest emphasizing mechanisms like the “escape from awareness" theory and its potential to bridge cognitive rigidity and emotional distress with binge behaviours.
Response 24: Thank you. We have followed your advice and have expanded on application of the theory to the cognitive rigidity section, see: “Cognitive rigidity in BED may contribute to the persistence of maladaptive eating patterns by hindering their ability to engage in adaptive problem solving [61] such as modifying established habits and routines surrounding food and eating. Particularly compelling is the “escape from awareness” theory [98], which provides a nuanced framework for understanding binge eating behaviours. This theoretical approach proposes that binge eating emerges as a complex coping mechanism, whereby individuals attempt to draw attention away from emotional distress by narrowing their cognitive focus to the immediate environment (food) rather than confronting aversive self-perceptions [99]. Consequently, individuals with BED may find it challenging to shift their focus away from food-related stimuli, resulting in increased preoccupation with food, heightened cravings, and ultimately, binge eating episodes.”
We also expanded emotion dysregulation section based on that theory, see “The aforementioned "escape from awareness" theory offers a nuanced framework for understanding this relationship, proposing that binge eating serves as a cognitive mechanism to avoid confronting overwhelming emotional experiences [98,99]. Specifically, individuals with BED may use food and binge eating as a strategic method of emotional avoidance, narrowing their cognitive focus to the immediate sensory experience of eating. This process is believed to block out emotional distress, providing temporary psychological relief [98]. Negative emotions, coupled with an inability to effectively process these feelings, may trigger binge eating as a maladaptive form of emotional regulation - a means of escaping awareness of underlying psychological discomfort [98,106]. The theory explains why individuals often report feeling "numb" or "disconnected" during binge episodes, highlighting binge eating not as a simple lack of self-control, but as a complex coping mechanism for managing emotional distress. Importantly, Bodell et al.’s (2019) [107] research suggests that emotional processing difficulties in BED are dynamic, context-dependent processes that can fluctuate over time and potentially be modified through targeted, momentary interventions.”
Comment 25: Including longitudinal findings or proposing hypotheses about the stability of neurocognitive impairments would add depth to the MS.
Response 25: We have now integrated longitudinal research support for each section. For cognitive flexibility, see:
“Notably, research suggests that cognitive rigidity is not merely a static trait but a dynamic process that can evolve over time. For instance, Messer et al. (2024) [96] demonstrated a significant longitudinal pathway where higher shape and weight overvaluation predicted increased inflexible adherence to food rules, which subsequently predicted increased binge eating symptoms over a 6-month period. Complementing these findings, early childhood studies by Steegers et al. (2021) [97] revealed that set-shifting difficulties are observable even before the typical onset of EDs.”
For impulsivity see:
“Longitudinal research provides critical insights into the developmental trajectory of impulsivity in EDs. For instance, Evans et al. (2019) [104] demonstrated that higher levels of impulsivity predicted the development of disordered eating attitudes over time, suggesting impulsivity may be a dynamic risk factor rather than a static trait. Pearson et al. (2015) [105] further illuminated this relationship, revealing that negative urgency and negative affect in children can predict early binge eating engagement through distinct pathways.”
For emotional processing see:
“Longitudinal research provides crucial insights into the dynamic nature of these emotional processing difficulties, revealing that emotion regulation is not a static trait but a fluid process that interacts with ED symptoms [107].”
Comment 26: Highlighting implications for therapeutic interventions targeting these cognitive-emotional links is needed.
Response 26: Thank you. We have now integrated suggestions for interventions.
For cognitive rigidity (though this also applies to the below section on emotional dysregulation), see “Specific intervention strategies might include cognitive rigidity and meta-cognitive training, which helps individuals recognise and modify rigid thinking patterns [100], and acceptance-based therapies that teach individuals to experience emotions without resorting to binge eating as an avoidance mechanism [82].”
For impulsivity, see “Future research should continue to explore the longitudinal pathways and potential intervention points in these developmental trajectories.”
For emotional dysregulation, see “Importantly, Bodell et al.’s (2019) [107] research suggests that emotional processing difficulties in BED are dynamic, context-dependent processes that can fluctuate over time and potentially be modified through targeted, momentary interventions.”
OSFED
Comment 27: The section fails to address the unique neurocognitive profiles of subtypes like atypical anorexia nervosa (AAN) and purging disorder (PD) thoroughly.
Response 27: Thank you for this comment. As outlined in the review, there are currently hardly any studies on neurocognitive profiles for the OSFED subtypes, including AAN and PD. We have outlined the research that currently exists and have highlighted the need for further research in the area.
Comment 28: Neuroimaging findings are described but do not discuss how they translate to functional or clinical outcomes.
Response 28: Thank you for this suggestion. We have now added the following sections to discuss how these neuroimaging findings translate to functional or clinical outcomes.
The following paragraphs were added to explain this further:
“6.1.1. Translation from neuroimaging findings to neurocognitive processes
The observed hyperactivation in reward, attention, and visual processing regions, coupled with hypo-activation in cognitive control areas, underscores a potential vulnerability in individuals with OSFED to food-related stimuli, potentially contributing to disordered eating behaviours. Clinically, these findings suggest that interventions targeting cognitive control processes, such as CRT [31], could help enhance regulatory responses to food cues in OSFED.
Similarly, the heightened prefrontal and cerebellar activity observed in response to food images, alongside obsessive-compulsive tendencies, may indicate that treatments addressing obsessive-compulsive symptoms, such as exposure and response prevention (ERP) [115], could be beneficial for certain OSFED presentations.
Finally, alterations in reward-related neural circuitry associated with binge eating symptoms in the OSFED subtypes displaying these symptoms, may suggest that approaches aimed at modulating reward sensitivity - such as mindfulness-based interventions [116] or reward restructuring therapies [117]) - could mitigate binge-eating behaviours by helping individuals develop healthier reward mechanisms.
Future research should focus on validating these neuroimaging findings in larger, more diverse OSFED samples to refine our understanding of these neural mechanisms and their functional implications. Upcoming research should also further explore OSFED subtype-specific neurocognitive traits and compare them to the more-established profiles observed in AN, BN and BED to enhance diagnostic precision and tailored interventions.
Comment 29: The authors should expand on subtype-specific neurocognitive traits and how they compare to established EDs.
Response 29: We have added the following sentence to the future research section for OSFED: “Upcoming research should also further explore OSFED subtype-specific neurocognitive traits and compare them to the more-established profiles observed in AN, BN and BED to enhance diagnostic precision and tailored interventions.”
Comment 30: Critically appraise the limited sample sizes and methodological heterogeneity in neuroimaging studies.
Response 30: We have now added this sentence to highlight these limitations further. “It should be noted though, while promising, these findings must be interpreted cautiously due to the limited sample sizes and methodological heterogeneity across neuroimaging studies, which constrain the generalizability and robustness of current conclusions.”
Comment 31: The authors should discuss practical implications for developing diagnostic tools and tailored treatments based on these findings.
Response 31: Thank you for this comment. We have addressed the practical implications in detail, as outlined in our response to Comment 28. In Section 8.1 (Clinical Implications – Classification), we emphasize the importance of incorporating neurocognitive factors into diagnostic assessments of eating disorders. This section also includes a detailed discussion on treatment implications. However, regarding OSFED, it is currently premature to provide specific commentary on these aspects, as research on the neurocognitive factors associated with OSFED remains in its early stages
Higher weight
Comment 32: The section overemphasizes cognitive flexibility and impulsivity while neglecting broader neurocognitive and emotional factors.
Response 32: Thank you for the reviewer’s feedback. In response, we have added several additional sections that focus on broader neurocognitive and emotional factors, as well as the implications of emotion regulation in individuals with HW.
Comment 33: The bidirectional relationship between HW and neurocognition is mentioned but needs to be adequately explored.
Response 33: Please see the section on summary of bidirectional relationship observed in a longitudinal study contrasted with previous research on bidirectionality.
We have added this section to explain the bi-directional nature in more detail:” For instance, in a longitudinal study spanning 9 years, HW negatively correlated with episodic memory without any changes in executive function [135]. However, the same study found that improved executive function over time corresponded to a decline in HW. Other research has shown that executive function deficits dysregulate eating behaviours and impair physical activity functioning as risk factors for HW [136]. These findings highlight the critical role of neurocognitive processes in shaping the risk and maintenance of HW, offering potential targets for intervention.”
Comment 34: It should be discussed how external factors, such as stigma and psychological stress, might mediate the relationship between HW and neurocognitive deficits.
Response 34: We have now described further the links between psychological stress and the impact of adaptive coping strategies of overeating in response to stress.
The following information was added to describe this in more detail:” It should also be noted that psychological distress and emotional regulation difficulties are key risk factors for HW. Overeating often emerges as an affective response to anxiety, sadness, or stress, reflecting maladaptive coping strategies when individuals struggle to regulate emotions [145]. Impaired self-regulation and ruminative thinking can drive emotional eating, where food is used to manage negative emotions. The “escape from awareness” theory [98] suggests that individuals overeat to avoid distressing thoughts or aversive self-awareness by focusing on immediate sensory experiences, such as the taste of food. Physiologically, chronic stress and abnormal cortisol secretion can stimulate appetite, promoting cravings for high-calorie, fatty, and sweet foods by activating brain reward systems and suppressing stress-response pathways [145,146]. Additionally, emotional regulation challenges may contribute to other HW risk factors, such as sedentary behaviour and sleep disturbances, further exacerbating weight gain [146].”
Comment 35: Please provide a deeper examination of the interplay between HW-specific emotional regulation issues and decision-making impairments.
Response 35: Research around emotional regulation challenges and maladaptive coping strategies, particularly the physiological basis of dysregulated cortisol has been discussed. Please see response provided to comment 34 above.
Comment 36: Mentioning interventions that consider physiological and psychosocial factors influencing HW neurocognition must be included.
Response 36: Thank you for this comment. We now discuss key interventions that target deficits in psychosocial and physiological factors influencing HW.
The following information is included in the treatment implication section:” Finally, in HW populations, impulsivity, reward processing difficulties, and emotional processing difficulties may perpetuate cycles of overeating and negatively impact body image. Mindfulness-based exercises can enhance focus and control over food-related impulses [159], while strategies from acceptance and commitment therapy and DBT [160] can help individuals reframe their relationship with food. Acceptance and commitment therapy encourages acceptance of cravings without acting on them, fostering healthier responses to food cues and reducing binge eating behaviour [161]. Together, these interventions can foster a more adaptive relationship with food, reduce the impact of impulsivity, reward processing difficulties, and emotional biases, and enhance overall well-being.”
Clinical implications
Comment 37: The discussion is weighted heavily towards traditional interventions like CRT and emotion regulation training, with insufficient focus on emerging therapies.
Response 37: Both traditional interventions, such as cognitive remediation therapy (CRT) and emotion regulation training, and emerging therapies like mindfulness, mental appraisal methods, and acceptance and commitment therapy (ACT), are described. Based on reviewers' feedback, we have integrated more detailed and specific treatment implications tailored to each ED subtype and HW, focusing on the relevant neurocognitive domains.
Comment 38: The section needs to consider the proposed treatments' feasibility, scalability, and cultural adaptability.
Response 38: Thanks for this comment. We have now added the following section to the clinical implications section to consider these aspects.
The following sentence was added:” It needs to be outlined though that the practical implementation of these treatments requires careful consideration of their feasibility, scalability, and cultural adaptability. CRT and its adaptations (e.g., CREST and TCRT) can be delivered in both individual and group formats, increasing scalability [162]. Incorporating digital platforms or app-based interventions may further enhance accessibility, particularly for remote or underserved populations. Emotion regulation therapies, such as DBT and its adaptations (e.g., DBT-BED), have demonstrated efficacy [158], but their intensive training requirements for clinicians could pose barriers to broader implementation. Simplified versions or modular approaches might improve scalability.”
Comment 39: Critique the effectiveness of existing interventions like CRT and propose refinements based on the review findings.
Response 39: This has already been included – we have outlined the emergence of CREST that includes both CRT and emotional training.
“In AN, potential deficits in cognitive flexibility, attentional biases, and emotional processing difficulties reveal specific targets for therapy. CRT (e.g., [31]) has been developed to help patients improve cognitive flexibility and central coherence, in addition to preparing patients to participate in other therapies [150]].
However, a recent meta-analysis of randomized controlled trials on CRT for AN showed no significant improvement in central coherence over control treatments at the end of treatment, but this finding was based on only three studies [151]. Cognitive Remediation and Emotion Skills Training (CREST) builds on CRT by integrating cognitive training with emotion skills training. This combination has shown promising results in AN patients, with significant improvements in social anhedonia, emotional labelling, and patients' confidence in their capacity for change [152].”
Comment 40: Recommendation of integrated neurocognitive training into current treatment paradigms for underrepresented populations (e.g., HW, OSFED) will benefit the MS depth.
Response 40: We already outlined this in the implication section previously. The information can be found in the following sections:
“For OSFED, the diverse presentations necessitate flexible and integrative interventions, as research on its classification, subtypes, and effective treatments is still in its infancy. This diagnostic category includes various symptom patterns that do not fully align with the other EDs, requiring approaches that address both the cognitive and emotional challenges specific to each presentation in addition to ED symptoms [108,109]. Combining elements from CRT/CREST, broader emotion regulation training, and attentional bias modification may effectively target these unique needs. As our understanding of OSFED develops, refining such targeted interventions will be essential for enhancing treatment outcomes and providing tailored support.”
“Finally, in HW populations, impulsivity, reward processing difficulties, and emotional processing difficulties may perpetuate cycles of overeating and negatively impact body image. Mindfulness-based exercises can enhance focus and control over food-related impulses [159], while strategies from acceptance and commitment therapy and DBT [160] can help individuals reframe their relationship with food. Acceptance and commitment therapy encourages acceptance of cravings without acting on them, fostering healthier responses to food cues and reducing binge eating behaviour [161]. Together, these interventions can foster a more adaptive relationship with food, reduce the impact of impulsivity, reward processing difficulties, and emotional biases, and enhance overall well-being”
Limitations
Comment 41: The limitations section recognizes critical gaps but needs to include a more profound critique of the studies' methodological and theoretical shortcomings.
Response 41: Thank you for this suggestion. We have now rewritten the limitation section. The sections addressing the methodological and theoretical limitations now read as:
“There are also various limitations within the reviewed studies that need to be considered. First, the study of neurocognitive profiles in EDs and HW populations is constrained by several methodological limitations. A significant challenge lies in the diversity of cognitive assessment tools utilized across studies. This variability in tasks designed to evaluate key domains such as cognitive flexibility, attentional bias, and emotion regulation complicates cross-study comparisons and hinders the identification of shared or disorder-specific impairments. For instance, while some studies use standardized neurocognitive tests, others employ experimental paradigms or self-report measures, each with varying levels of reliability and ecological validity.
Second, many studies lack uniformity in the operational definitions and measurement of neurocognitive constructs, leading to inconsistent results. Small and non-representative sample sizes are another limitation, often reducing the generalizability of findings to broader clinical populations. Few studies stratify their samples by important demographic or clinical factors, such as age, gender, comorbidities, or duration of illness/condition, which may influence neurocognitive functioning.
Third, theoretical limitations also warrant consideration. Many studies fail to clearly differentiate between trait-like neurocognitive deficits that may predispose individuals to EDs and HW conditions and state-like impairments that emerge as consequences of these disorders or associated factors such as malnutrition or weight changes. This lack of theoretical clarity complicates the interpretation of findings and the development of targeted interventions. Furthermore, it remains unclear which cognitive factors are inherent traits, which exist premorbidly, which are scar effects, and which are symptoms of the ED illness and HW, underscoring the need for more nuanced research in this area.
Fourth, the reliance on cross-sectional studies restricts understanding of how neurocognitive deficits develop and progress across different stages of ED, its recovery, and HW. The limited amount of longitudinal research available on EDs and cognition restricts insights into whether neurocognitive impairments endure over time or improve with treatmentand recovery. For example, longitudinal tracking of cognitive flexibility in AN could help clarify whether such deficits improve with weight restoration and recovery or if they persist as chronic features of the disorder. Similarly, observing attentional biases in BN and BED across treatment phases could reveal whether these biases diminish as part of recovery or if they remain stable, potentially contributing to relapse risk.”
Comment 42: The reliance on cross-sectional data is mentioned but needs to be elaborated on sufficiently to highlight its impact on findings.
Response 42: Please see response provided to comment 41 above.
Comment 43: The authors should discuss how the diversity in assessment tools (e.g., self-report vs. experimental) and small sample sizes limit generalizability.
Response 43: Please see response provided to comment 41 above.
Comment 44: Exploring the implications of omitting ARFID and childhood-onset EDs for the review’s conclusions is required.
Response 44: Thank you for raising tis point. We have now added the following sentences to address the implications of omitting ARFID for this review:” Omitting ARFID from the review limits the comprehensiveness of understanding neurocognitive factors across the full spectrum of EDs. It overlooks unique developmental and neurocognitive challenges associated with early-onset disorders, such as sensory sensitivities and anxiety-related behaviours, which differ from those observed in other EDs (4,5). This omission hinders the ability to develop targeted interventions and to inform clinical practice for ARFID populations.”
Comment 45: The authors can call for greater methodological rigour in future studies, including harmonizing cognitive task designs and diagnostic criteria.
Response 45: Please see response provided to comment 41 above.
Future directions
Comment 46: Recommendations for future research are general and lack specific testable hypotheses or methodological frameworks.
Response 46: We have now rephrased this whole section and included specific testable hypotheses or methodological frameworks. See for instance example provided below: “Key challenges include maintaining participant engagement over extended periods, managing attrition rates, and the need for sophisticated methodologies to assess cognitive changes reliably. Specific testable hypotheses for future studies should include: Are observed neurocognitive deficits in these disorders temporary and associated with the current illness stage or severity, or are they stable traits that persist beyond weight restoration? It would also be important to further assess how specific neurocognitive traits predict recovery trajectories or relapse risks in EDs and HW. Future studies could for instance assess the hypothesis: “Do deficits in cognitive flexibility and impulse control at the start of treatment predict poorer recovery trajectories in AN patients?” Such concrete research questions will help guide future studies in identifying predictors of treatment success and relapse, thus informing more personalized and effective intervention strategies.”
Comment 47: The emphasis on longitudinal studies is valid, but the challenges of such research need to be addressed.
Response 47: We have now added the following information in relation to longitudinal studies in this section: “Future research should focus on refining our understanding of neurocognitive profiles in AN, BN, BED, OSFED and HW through detailed longitudinal studies. Longitudinal studies are needed to track cognitive changes over time and capture the dynamic nature of these disorders and their recovery phases. However, the challenges of such research need to be addressed. Key challenges include maintaining participant engagement over extended periods, managing attrition rates, and the need for sophisticated methodologies to assess cognitive changes reliably.”
Comment 48: Proposing concrete research questions, such as how specific neurocognitive traits predict recovery trajectories or relapse risks, can be more relevant to readers.
Response 48: We have added the following information to address this: “It would also be important to further assess how specific neurocognitive traits predict recovery trajectories or relapse risks in EDs and HW. Future studies could for instance assess the hypothesis: “Do deficits in cognitive flexibility and impulse control at the start of treatment predict poorer recovery trajectories in AN patients?” Such concrete research questions will help guide future studies in identifying predictors of treatment success and relapse, thus informing more personalized and effective intervention strategies.”
Comment 49: The authors can discuss practical methodologies, such as combining neuroimaging with behavioural data or integrating biomarkers.
Response 49: We have added a section that discusses how different sources of data courses could be combined in future research.
We have added the following section for this: “Incorporating neuroimaging data can further refine this understanding. Neuroimaging studies, such as functional magnetic resonance imaging (fMRI), can elucidate changes in brain activity related to cognitive flexibility, reward processing, and other executive functions in these populations. Integrating neuroimaging with biomarkers could lead to a more comprehensive understanding of the biological underpinnings of cognitive impairments in these populations and help in designing targeted interventions that address both nutritional and cognitive deficits effectively.”
Comment 50: Addressing the need for interdisciplinary collaborations to investigate the neurobiological underpinnings of EDs and HW populations will enhance the rigidity of this sub-section.
Response 50: Thank you for raising this comment. We have now added the following paragraph to outline the need to interdisciplinary collaborations:
“Finally, there is a need for further interdisciplinary collaborations to investigate neurocognitive profiles in EDs and HW populations. These collaborations will enable the integration of diverse expertise, including psychology, neuroscience, nutrition, and clinical treatment strategies, to develop a holistic understanding of the complex interplay between neurocognitive functioning, physiological markers, and behavioural data. Such an approach will be essential in advancing personalized treatment approaches that are responsive to the unique needs of individuals with EDs and HW.”
Conclusions
Comment 51: The conclusion reiterates findings without providing actionable insights or a strong narrative linking all sections.
Response 51: We have now rewritten the whole Conclusion section and hope that the revised version now provides a stronger narrative linking all the sections.
Comment 52: The need for individualized treatment approaches is emphasized but not grounded in specific examples from the review.
Response 52: We have now also added a few sentences for the individualized treatment approaches for the different ED subtypes and HW. The following information was provided to outline this:
“It highlights the significance of poor cognitive flexibility and attentional biases in AN, underscoring the need for tailored interventions to address these specific cognitive patterns, such as CRT/CREST and CBT strategies. For BN and BED, the review discusses how poor cognitive flexibility, emotional processing difficulties, and heightened reward sensitivity contribute to binge-eating and purging behaviours, suggesting the need for therapeutic approaches that manage these complex relationships such as DBT-BED. Finally, individuals with HW face neurocognitive challenges like impaired decision-making and executive function, further emphasizing the need for individualized, transdiagnostic treatment strategies. Given that research on OSFED subtypes is still in its infancy, conclusions regarding OSFED remain preliminary and highlight the need for further investigation to better understand these subtypes and their unique cognitive challenges.”
Comment 53: A compelling argument for integrating neurocognitive findings into ED classification systems and clinical frameworks is needed.
Response 53: We have also addressed a section on this. The new information states that:” Future research should prioritize longitudinal studies to explore the evolution of these neurocognitive profiles over time, conduct further neuroimaging, consider biomarkers, and collaborate with other fields. Such research will ensure the development of effective, tailored interventions that improve recovery outcomes across diverse ED and HW populations.”
Comment 54: Emphasizing the review’s unique contributions while acknowledging its limitations should be introduced more explicitly.
Response 54: We have added a few sentences on this:
For the unique contribution we included the following introductory sentence: “To conclude, the unique contributions of the current review include its wide coverage of the most up-to-date literature on neurocognitive factors across EDs and HW, providing a comprehensive synthesis of the existing research in this area.”
For the limitations we added the following sentences: “Given that research on OSFED subtypes is still in its infancy, conclusions regarding OSFED remain preliminary and highlight the need for further investigation to better understand these subtypes and their unique cognitive challenges.
General Criticisms
Comment 55: Many sections repeat similar points (e.g., BMI’s role in neurocognitive deficits), which could be consolidated.
Response 55: We have rewritten several sections of the review to consolidate the information presented.
Comment 56: Findings across disorders are presented in isolation, missing an opportunity to identify transdiagnostic patterns.
Response 56: We do refer to transdiagnostic patterns at various places throughout the review. Below are some of these examples:
“We found that some neurocognitive characteristics, such as cognitive rigidity and dysregulated reward processing, appear transdiagnostic, spanning multiple ED subtypes and HW populations. At the same time, we also revealed neurocognitive features specific to ED subtypes and HW”
“By identifying transdiagnostic factors, such as cognitive rigidity and reward processing, alongside ED subtype/HW-specific vulnerabilities, researchers and clinicians can develop more nuanced, evidence-based interventions that address the core mechanisms driving disordered eating behaviours.”
“Importantly, some of these neurocognitive factors, such as cognitive flexibility, appear to be transdiagnostic, meaning they may be present across multiple ED subtypes.”
Comment 57: OSFED and HW populations are discussed superficially compared to AN and BN, reflecting a bias in coverage.
Response 57: The discussion of OSFED was intentionally more limited due to the current state of research and available literature on these groups. Given the extensive research on AN, BN and more recently also BED, it was necessary to prioritize these areas to provide a more in-depth analysis, reflecting the breadth and depth of knowledge available. OSFED, while important, were covered more superficially because there is still limited research specific to these groups. This approach does not reflect bias; rather, it reflects a need to focus on well-established areas of study within the scope of our paper.
As regards to HW populations, we have now expanded this section to provide a more balanced coverage in relation to other ED sections. This adjustment reflects our effort to address the unique characteristics and implications of HW more comprehensively. However, we did not replicate the exact structure or depth of analysis used for ED sections, as the aim was to focus on distinct comparisons and to align with the available research evidence without direct comparisons to other ED categories.
Comment 58: While the paper is well-referenced, some sections rely on excessive citations without critically evaluating sources.
Response 58: Thank you this raising this point. We included a broad range of references due to the constrained word limit and the need to provide a comprehensive overview of the neurocognitive processes and treatment implications across different eating disorder subtypes and higher weight. This wide-ranging citation strategy ensured that all relevant aspects of the topic were addressed, allowing us to cover key studies and their findings. While this meant we couldn’t critically evaluate each source in depth, it was necessary to present a foundational understanding of the evidence base, providing a balanced view of the complexity involved without overwhelming the reader.
Reviewer 3 Report
Comments and Suggestions for Authors
The manuscript submitted by Krug et al. provides relevant data on the topic and it can be considered for publication in Nutrients after these revisions:
In the abstract, you need to provide some details about the searched databases, inclusion/exclusion criteria, and future perspectives.
I suggest you include a deeper literature review in the introductory section. The readers need to understand the novelty and justification of the study you are conducting. Some figures and tables are encouraged to be included here.
Despite being a narrative review, a Methods section is missing. See the example of the section 2 of this published paper: https://www.mdpi.com/2304-8158/10/6/1175
Improve your subsection 4.3. with further discussions.
The main drawback of your work is the lack of figures and tables to summarize the discussed studies.
Your conclusion can also be improved, and more data should be provided about what should be done next based on your results.
Author Response
The manuscript submitted by Krug et al. provides relevant data on the topic and it can be considered for publication in Nutrients after these revisions:
Comment 1: In the abstract, you need to provide some details about the searched databases, inclusion/exclusion criteria, and future perspectives.
Response 1: Thank you for suggesting this. We have now added this information in the abstract.
“A search of major databases that prioritized meta-analyses and recent publications (last 10 years) was conducted. Using comprehensive search terms related to EDs, HW, and neurocognition, eligible studies focused on human neurocognitive outcomes (e.g., cognitive flexibility, attentional bias etc.) and were published in English.”
Comment 2: I suggest you include a deeper literature review in the introductory section. The readers need to understand the novelty and justification of the study you are conducting. Some figures and tables are encouraged to be included here.
Response 2: We have expanded the introduction and strengthened the rationale for the review, ensuring these additions are presented succinctly to maintain clarity and focus. Recognizing the comprehensive nature of this review and its increased length following revisions, we have carefully integrated these elements to balance depth and conciseness.
Comment 3: Despite being a narrative review, a Methods section is missing. See the example of the section 2 of this published paper: https://www.mdpi.com/2304-8158/10/6/1175
Response 3: Thank you for this suggestion and for outlining the link. That has been very helpful to follow. We have now added a Method section in Section 2.
Comment 4: Improve your subsection 4.3. with further discussions.
Response 4: We have rewritten several sections of the review, including 4.3.
Comment 5: The main drawback of your work is the lack of figures and tables to summarize the discussed studies.
Response 5: Thank you for this comment. We have now added a table that summarizes the main findings of the review
Comment 6: Your conclusion can also be improved, and more data should be provided about what should be done next based on your results.
Response 6: Thank you for this comment. Reviewer 1 has raised a similar problem with our conclusion. We have now rephrased this section.